 SciPost Phys. Lect. Notes 71 (2023)

# The WIMP paradigm: Theme and variations

### Jonathan L. Feng

Department of Physics and Astronomy, University of California, Irvine, CA 92697, USA

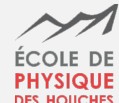

*Part of the Dark Matter*
*Session 118 of the Les Houches School, July 2021*
*published in the Les Houches Lecture Notes Series*

## Abstract

WIMPs, weakly-interacting massive particles, have been leading candidates for particle dark matter for decades, and they remain a viable and highly motivated possibility. In these lectures, I describe the basic motivations for WIMPs, beginning with the WIMP miracle and its under-appreciated cousin, the discrete WIMP miracle. I then give an overview of some of the basic features of WIMPs and how to find them. These lectures conclude with some variations on the WIMP theme that have by now become significant topics in their own right and illustrate the richness of the WIMP paradigm.

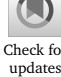

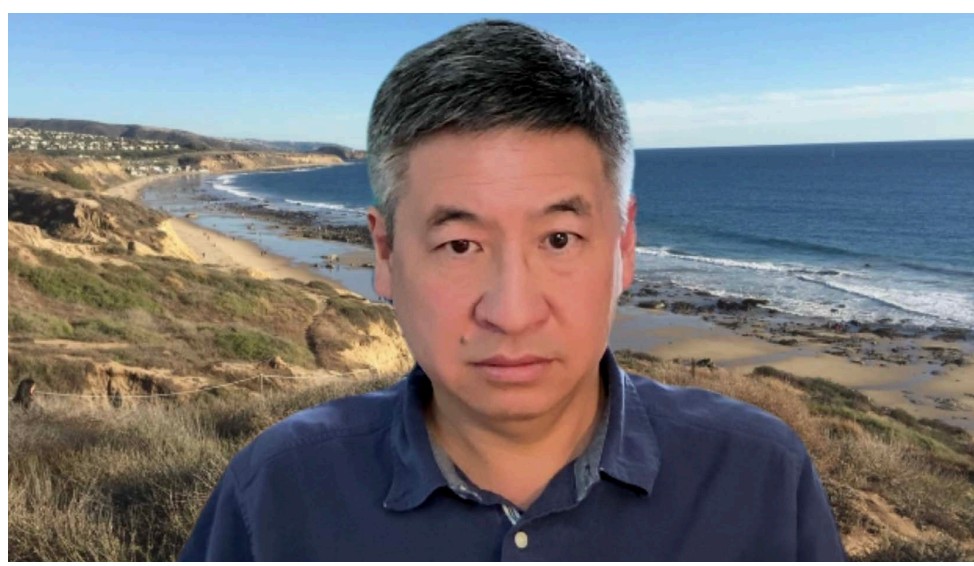

## 1 Advice

The field lost one of its giants just a few days before these lectures were given. Steve Weinberg (Fig. 1), had an enormous influence on particle physics and particle physicists. For me, it began early: wandering around the physics department as an undergraduate, I saw a poster for his 1987 Loeb Lectures, and a few days later, I attended them. These were the first professional physics talks I ever heard, and they remain a wonderful memory. Little did I know then how unusual this first exposure was or that one could attend many more physics talks for decades without hearing anything similarly interesting about the cosmological constant problem.

Weinberg wrote about many things, both within physics and beyond physics. I respected him too much to take any of it lightly, and not all of it was equally enlightening to me, notably his views on science and faith. But there was an awful lot to learn from and admire. In 2003, in a 1-page article in *Nature* [1], he shared the following "Four Golden Lessons":
- No one knows everything, and you don't have to.
- Go for the messes—that's where the action is.
- Forgive yourself for wasting time [working on the wrong questions].
- Learn something about the history of your own branch of science.

These are pearls of wisdom, and these lectures will be a success if they encourage a few students to follow this excellent advice.

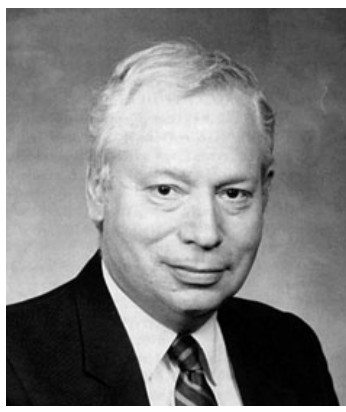

Figure 1: Steve Weinberg (1933-2021).

## 2 Introduction

Figure 2 is a figure I prepared a few years ago to summarize the landscape of particle dark matter candidates. As with any diagram of this sort, it glosses over many details. Remarkably, it was still too controversial to be approved by a committee—see the considerably blander version that made it into Ref. [2]! The diagram does, however, get a few main points across. First, it is clear that there are many candidates (blue), many interesting anomalies that motivate them (red), and many experimental methods that can be used to search for them (green). But perhaps most striking is the mass range: $10^{-21}$ eV at the low end to many solar masses at the high end, a dynamic range of almost $10^{100}$. The variety of possible dark matter candidates is truly vast, as is the range of their possible masses and interaction strengths.

WIMPs, weakly-interacting massive particles [3], are only one species in this zoo of dark matter candidates. There is no precise definition of "WIMP," but in these lectures, we will take WIMPs to be particles with masses in the $\sim 10$ GeV to 10 TeV range that interact through the weak interactions of the standard model (SM). As highlighted in Fig. 2, the WIMP mass range covers only a small subset of the possible masses. Despite this, WIMPs have commanded the lion's share of the attention of dark matter theorists and experimentalists in the last several decades, and the WIMP paradigm is essential background for almost any discussion of particle dark matter. It would be hard to imagine giving lectures on dark matter production without talking about WIMP freezeout, dark matter direct detection without talking about WIMP direct detection, dark matter indirect detection without talking about WIMP indirect detection, or dark matter at accelerators without talking about WIMP searches at colliders.

Why is this? The goal of these lectures is to answer this question. But in this brief introduction, we can give two quick answers:

- **The WIMP Miracle.** WIMPs are motivated by particle theory, as they appear in many beyond-the-SM (BSM) models designed to shed light on particle physics puzzles. WIMPs are also motivated by particle experiment, in the sense that they are at the current frontier, not excluded by existing bounds, but detectable at near future experiments. Last, WIMPs are motivated by cosmology: as we will see in Sec. 3.2, assuming the simple production mechanism of thermal freezeout, WIMPs are produced with the right relic density to be dark matter. This remarkable triple coincidence, that particle theory, particle experiment, and cosmology all motivate WIMPs, that is, particles with couplings $g \sim 1$ and masses $m \sim 10$ GeV $- 10$ TeV, is sometimes called the *WIMP miracle* [4, 5]. It is illustrated in Fig. 3. That studies of nature at both the smallest and largest lengths scales should point to particles with the same properties is extremely interesting and

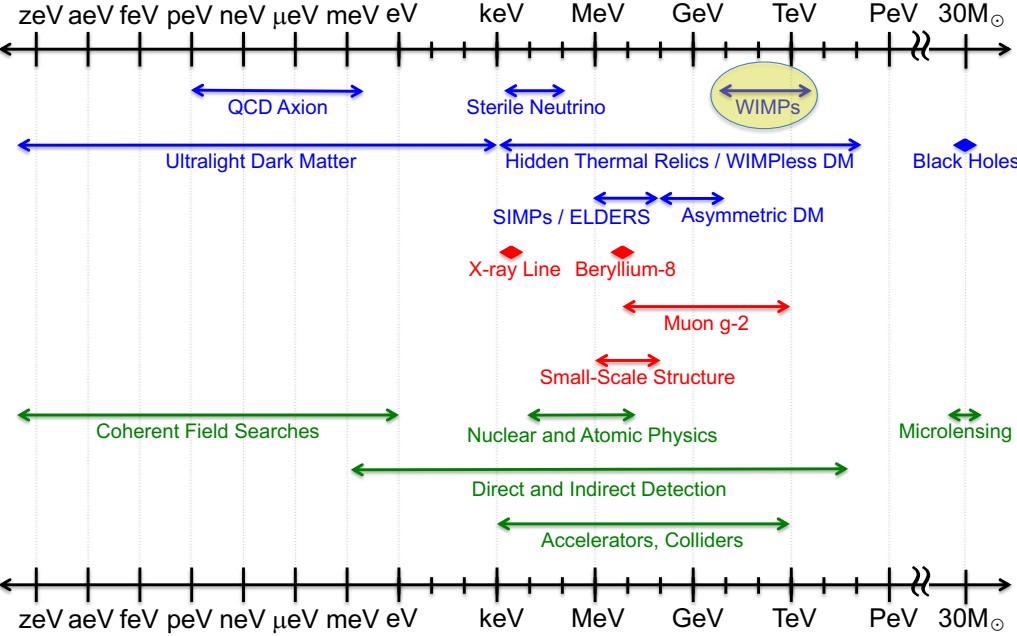

Figure 2: The landscape of particle dark matter candidates (blue), spanning almost a 100 orders of magnitude in mass, along with interesting anomalies (red) and detection methods (green). WIMPs are highlighted.

makes WIMPs highly motivated among the many dark matter candidates.[1]

**Dark Matter Complementarity.** WIMP dark matter has many implications for a diverse group of search experiments. This is illustrated in Fig. 4. As we will see in Sec. 3.2, if WIMPs are produced by thermal freezeout in the early universe, there is typically an effective DM-DM-SM-SM 4-point interaction. By viewing this interaction with the arrow of time running in various directions, this interaction implies dark matter annihilation now, dark matter scattering off normal matter, and the possibility of creating dark matter in the collisions of SM particles, leading to the possibility of discovering WIMP dark matter through indirect detection, direct detection, and collider searches, respectively. The fact that dark matter can be searched for in so many interesting and inter-related ways is known as *dark matter complementarity*. Note that, in many cases, the requirement that dark matter not be over-produced in the early universe typically implies lower bounds for all of these interactions, providing promising targets for a large variety of searches.

In these lectures, I will begin by discussing the most important motivations for WIMPs, including both the WIMP miracle and its under-appreciated cousin, the discrete WIMP miracle. I will then discuss the basic features of WIMPs and how to find them, considering WIMPs in supersymmetry (SUSY) as a concrete example. At the end, I will present some variations on the WIMP theme, where relatively minor tweaks lead to a wealth of qualitatively different features, illustrating the richness of the WIMP paradigm.

These lectures are written primarily for graduate students starting dark matter research, but it is hoped there will be something of interest for others as well. They will stress the basic ideas and order-of-magnitude estimates, and they will be short on computational details. For

---

[1]Note that, in addition to WIMPs, particle experiment and cosmology also motivate another class of particles: feebly-interacting light particles with the correct thermal relic density [6–8]. These have couplings $g \ll 1$ and masses $m \ll 10$ GeV, and so are not WIMPs, but they may be thought of as variations on the WIMP paradigm, and they will be discussed further in Sec. 6.4.

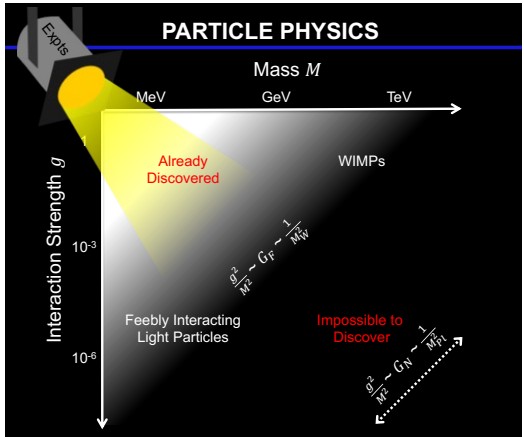
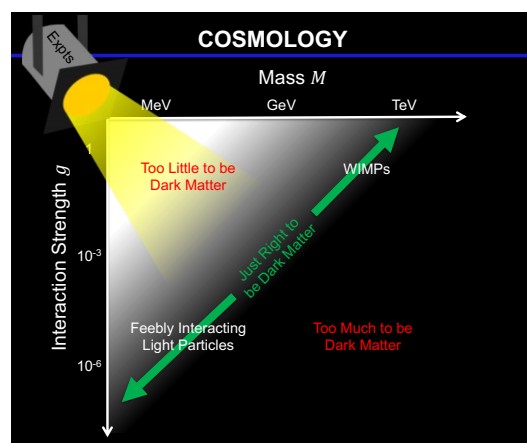

Figure 3: The new particle landscape in the (mass, coupling) plane from the perspectives of particle physics (left) and cosmology (right). On the left, WIMPs lie on the diagonal with $g^2/M^2 \sim G_F$, the Fermi constant, where many particle theories predict new particles, and which is also the current frontier of particle experiments. On the right, WIMPs lie on the same diagonal where particles are produced in thermal freezeout with the correct relic density to be dark matter. The triple coincidence of motivations from particle theory, particle experiment, and cosmology is known as the WIMP miracle.

more comprehensive treatments, see, e.g., Ref. [10], which serves as a source for these lectures, Refs. [11–14], and the other lectures at this school. And last, heeding Weinberg's fourth lesson, see Ref. [15] for an account of the history of dark matter, written by physicists for physicists.

# 3 Why WIMPs?

As noted above, WIMPs are among the most studied class of particle dark matter candidates because they arise naturally in many particle physics theories, have the correct cosmological properties, and have a breathtakingly diverse set of implications for observable phenomena. In this section, we discuss the first two of these motivations. We will return to the third in Sec. 5.

## 3.1 The Weak Scale

In the 1930's, Fermi introduced his constant in the study of nuclear beta decay. The value of the Fermi constant, $G_F \simeq 1.2 \times 10^{-5} \text{ GeV}^{-2}$, introduces a new energy scale in physics, the weak scale $m_{\text{weak}} \sim G_F^{-1/2} \sim 100$ GeV. We still do not understand the origin of this scale, but so far, every reasonable attempt to understand it has introduced new particles with masses around the weak scale.

What would it mean to understand the origin of the weak scale? A good first step would be to understand why it is so small compared to the Planck mass. We know of three fundamental constants: the speed of light $c$, Planck's constant $h$, and Newton's gravitational constant $G_N$. One combination of these has dimensions of mass, the Planck mass $M_{\text{Pl}} \equiv \sqrt{hc/G_N} \simeq 1.2 \times 10^{19}$ GeV. We therefore expect dimensionful parameters to be either 0, if enforced by a symmetry, or of the order of $M_{\text{Pl}}$. In the SM, however, electroweak symmetry is broken, but the weak scale is far below $M_{\text{Pl}}$.

This puzzle of the weak scale has, if anything, been heightened by the discovery in 2012 of

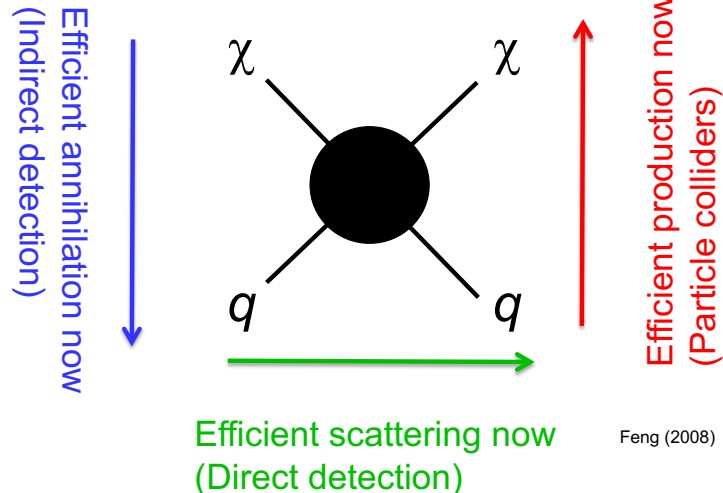

Figure 4: The complementarity of various WIMP dark matter detection methods [9]. The WIMP miracle requires efficient annihilation of WIMPs $\chi$ to SM particles $q$ in the early universe. This, in turn, typically implies efficient dark matter annihilation now, efficient dark matter scattering, and dark matter production, implying promising rates for indirect detection, direct detection, and collider searches, respectively.

the Higgs boson. With increasing precision, the Higgs boson appears to be a fundamental scalar with a mass of $m_h \simeq 125$ GeV. The discovery of the Higgs boson was a watershed moment in particle physics, in part because it showed that fundamental scalars, rather than being simply a pedagogical tool for quantum field theory courses or a fun toy for model builders, actually exist in nature.

Fundamental scalars are fundamentally different. Unlike the other particles of the SM, the mass of the Higgs boson receives quantum corrections that are quadratically dependent on the cutoff $\Lambda$, the scale at which new physics enters. This puzzle, the gauge hierarchy problem, also known as the naturalness or fine-tuning problem, has typically been taken as one of the leading motivations for new physics, and typically leads to the prediction of particles with mass $M \sim m_W$ and couplings $g \sim 1$; see, for example, Refs. [16–18].

On the cosmological side, there is, of course, also a strong motivation for new particles: the dark matter problem. Dark matter is known to have the following properties: it must be (1) gravitationally interacting, (2) not short-lived, (3) not hot, and (4) not baryonic. The first requirement is not much of a requirement—all particles gravitationally interact, even massless ones. But the remaining three requirements are sufficient to exclude all known SM particles from being dark matter, and so require new particles.

Given the need for new particles to address central problems in both particle physics and cosmology, it is tempting to hope for a single solution to both problems. This hope is, in fact, supported by the tantalizing numerical coincidence of the WIMP miracle, to which we now turn.

## 3.2 The WIMP Miracle

If WIMPs exist and are stable, they are naturally produced with a relic density consistent with that required of dark matter. This implies that particles that are motivated by attempts to understand the weak scale, as discussed in Sec. 3.1, a purely microphysical puzzle, are often excellent dark matter candidates.

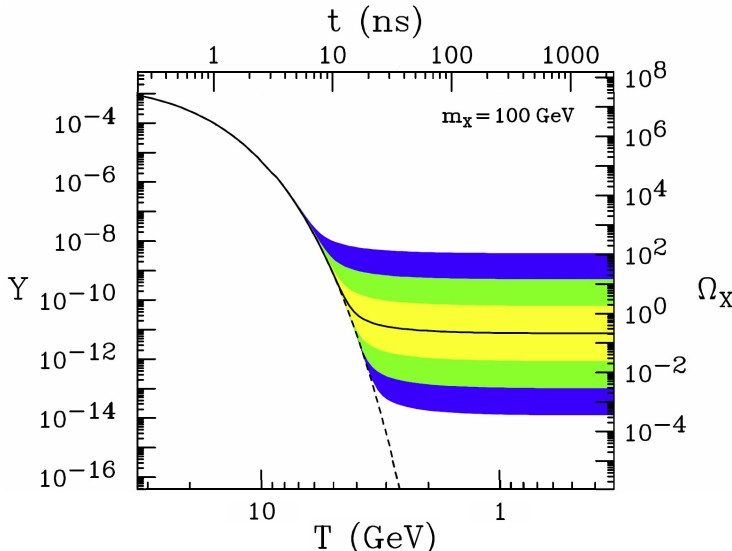

Figure 5: The comoving number density $Y \equiv n_X/s$, the ratio of number density to entropy density (left) and resulting thermal relic density (right) of a 100 GeV, $P$-wave annihilating dark matter particle as a function of temperature $T$ (bottom) and time $t$ (top). The solid contour is for an annihilation cross section that yields the correct relic density, and the shaded regions are for cross sections that differ by 10, $10^2$, and $10^3$ from this value. The dashed contour is the number density of a particle that remains in thermal equilibrium. From Ref. [10].

Dark matter may be produced in a simple and predictive manner as a thermal relic of the Big Bang [19–22]. The evolution of a thermal relic's number density is shown in Fig. 5. Initially the early universe is dense and hot, and all particles are in thermal equilibrium. The universe then cools to temperatures $T$ below the dark matter particle's mass $m_X$, and the number of dark matter particles becomes Boltzmann suppressed, dropping exponentially as $e^{-m_X/T}$. The number of dark matter particles would drop to zero, except that, in addition to cooling, the universe is also expanding. Eventually the universe becomes so large and the gas of dark matter particles becomes so dilute that they cannot find each other to annihilate. The dark matter particles then "freeze out," with their number asymptotically approaching a constant, their thermal relic density.

This process is described quantitatively by the Boltzmann equation

$$\frac{dn}{dt} = -3Hn - \langle \sigma_A v \rangle \left( n^2 - n_{\text{eq}}^2 \right), \tag{1}$$

where $n$ is the number density of the dark matter particle $X$, $H$ is the Hubble parameter, $\langle \sigma_A v \rangle$ is the thermally-averaged dark matter annihilation cross section, and $n_{\text{eq}}$ is the dark matter number density in thermal equilibrium. On the right-hand side of Eq. (1), the first term accounts for dilution from expansion. The $n^2$ term arises from processes $XX \to$ SM SM that destroy $X$ particles, where SM here denotes SM particles, and the $n_{\text{eq}}^2$ term arises from the reverse process SM SM $\to XX$, which creates $X$ particles.

The thermal relic density is best determined by solving the Boltzmann equation numerically. A rough analysis is highly instructive, however. Defining freezeout to be the time when the interaction rate is equal to the expansion rate, $n\langle \sigma_A v \rangle = H$, and assuming freezeout during the radiation-dominated era, we have

$$n_f \sim (m_X T_f)^{3/2} e^{-m_X/T_f} \sim \frac{T_f^2}{M_{\text{Pl}} \langle \sigma_A v \rangle}, \tag{2}$$

where the subscripts $f$ denote quantities at freezeout. The ratio $x_f \equiv m_X/T_f$ appears in the exponential. It is, therefore, highly insensitive to the dark matter's properties and may be considered a constant; a typical value is $x_f \sim 20$. The thermal relic density is, then,

$$\Omega_X = \frac{m_X n_0}{\rho_c} = \frac{m_X T_0^3}{\rho_c} \frac{n_0}{T_0^3} \sim \frac{m_X T_0^3}{\rho_c} \frac{n_f}{T_f^3} \sim \frac{x_f T_0^3}{\rho_c M_{\text{Pl}}} \langle \sigma_A v \rangle^{-1}, \tag{3}$$

where $\rho_c$ is the critical density, the subscripts 0 denote present day quantities, and we have assumed an adiabatically expanding universe. We see that the thermal relic density is insensitive to the dark matter mass $m_X$ and inversely proportional to the annihilation cross section $\langle \sigma_A v \rangle$.

Although $m_X$ does not enter $\Omega_X$ directly, in many theories it is the only mass scale that determines the annihilation cross section. On dimensional grounds, then, the cross section can be written

$$\sigma_A v = k \frac{g_{\text{weak}}^4}{16\pi^2 m_X^2} (1 \text{ or } v^2), \tag{4}$$

where the factor $v^2$ is absent or present for $S$- or $P$-wave annihilation, respectively, and terms higher-order in $v$ have been neglected. The constant $g_{\text{weak}} \simeq 0.65$ is the weak interaction gauge coupling, and $k$ parametrizes deviations from this estimate.

With this parametrization, given a choice of $k$, the relic density is determined as a function of $m_X$. The results are shown in Fig. 6. The width of the band comes from considering both $S$- and $P$-wave annihilation, and from letting $k$ vary from $\frac{1}{2}$ to 2. We see that a particle that makes up all of dark matter is predicted to have mass in the range $m_X \sim 100 \text{ GeV} - 1 \text{ TeV}$; a particle that makes up 10% of dark matter has mass $m_X \sim 30 \text{ GeV} - 300 \text{ GeV}$. This is the WIMP miracle: weak-scale particles with $\mathcal{O}(1)$ couplings freeze out with the desired relic density and make excellent dark matter candidates.

We have neglected many details here, and there are models for which $k$ lies outside our illustrative range, sometimes by as much as an order of magnitude or two. Nevertheless, the WIMP miracle implies that many models of particle physics easily provide viable dark matter candidates, and it is at present the strongest reason to expect that central problems in particle physics and astrophysics may in fact be related. Note that the WIMP miracle is a triple coincidence of motivations from particle theory, particle experiment, and cosmology. Even if one has no interest in BSM theories designed to explain the weak scale or considers the gauge hierarchy problem a purely aesthetic issue, the WIMP miracle independently provides a strong motivation for new particles at the weak scale which can be probed by particle experiments at the current frontier of energy or sensitivity.

The WIMP miracle has a number of interesting implications that are worth noting. First, WIMPs freeze out with $m/T \sim 20$, which corresponds to typical velocities of $v \sim \frac{1}{3}$ (in units of $c$). At freezeout, then, WIMPs were neither ultra-relativistic nor non-relativistic. This contrasts sharply with the non-relativistic velocities of WIMPs in our neighborhood now, which are $v \sim 10^{-3}$. This difference has been exploited to great effect in many different ways in the literature, and we will see an illustration of this when we discuss inelastic dark matter in Sec. 6.1. Second, as a minor corollary, this implies that, for a 100 GeV WIMP, freezeout occurs at temperatures of $T \sim 5 \text{ GeV}$ and times $t \sim \text{ns}$ after the Big Bang, not temperatures of $T \sim m_{\text{weak}}$ and times $t \sim \text{ps}$, as is sometimes assumed. Last, the freezeout we have been discussing is also known as chemical decoupling and is distinct from kinetic decoupling: after chemical decoupling, number changing processes become inefficient, whereas after kinetic decoupling, energy-changing processes become inefficient. After thermal freezeout, interactions that change the number of dark matter particles become negligible, but interactions that mediate energy exchange between dark matter and other particles may remain efficient.

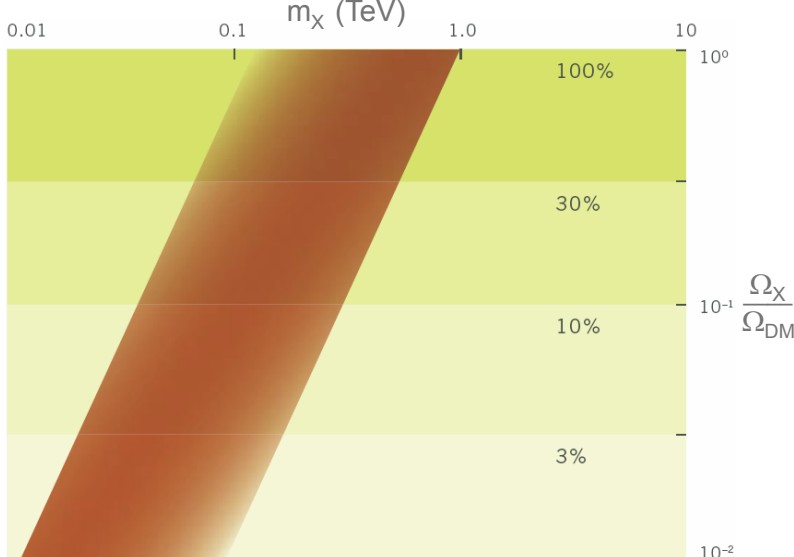

Figure 6: A band of natural values in the $(m_X, \Omega_X/\Omega_{\mathrm{DM}})$ plane for a thermal relic $X$, where $\Omega_{\mathrm{DM}} \simeq 0.23$ is the required total dark matter density. From Ref. [23].

## 3.3 The Discrete WIMP Miracle

The entire discussion of Sec. 3.2 assumes that the WIMP is stable. This might appear to be an unreasonable expectation; after all, all particles heavier than a GeV in the SM decay on time scales far shorter than the age of the universe. In fact, however, there are reasons to believe that if new particles exist at the weak scale, at least one of them should be stable. This is the cosmological legacy of LEP, the Large Electron-Positron Collider at CERN that ran from 1989-2000.

In many BSM models designed to address the gauge hierarchy problem, new particles are introduced that interact with the Higgs boson through couplings

$$h \longleftrightarrow \mathrm{NP\,NP}. \tag{5}$$

These contribute to the Higgs boson mass through diagrams shown in Fig. 7, and their masses are expected to be around $m_{\mathrm{weak}} \sim 10\,\mathrm{GeV} - \mathrm{TeV}$.

Unfortunately, these same new particles generically induce new interactions

$$\mathrm{SM\,SM} \rightarrow \mathrm{NP} \rightarrow \mathrm{SM\,SM}, \tag{6}$$

where SM and NP denote standard model and new particles, respectively, through the Feynman diagrams shown in Fig. 7. If the new particles are heavy, they cannot be produced directly, but their effects may nevertheless be seen as perturbations on the properties of SM particles. LEP, along with the Stanford Linear Collider, looked for the effects of these interactions and found none, constraining the mass scale of new particles to be above $\sim 1 - 10$ TeV, depending on the SM particles involved (see, *e.g.*, Ref. [24]).

These apparently conflicting demands may be reconciled if there is a conserved discrete parity under which all SM particles are even and all new particles are odd [25, 26]. Parity conservation then requires that all interactions involve an even number of new particles. Such a conservation law preserves the desired interactions of Eq. (5), but eliminates the problematic reactions of Eq. (6). As a side effect, the existence of a discrete parity implies that the lightest new particle cannot decay, as shown in Fig. 8. The lightest new particle is therefore stable, as required for dark matter. Note that pair annihilation of dark matter particles is still allowed.

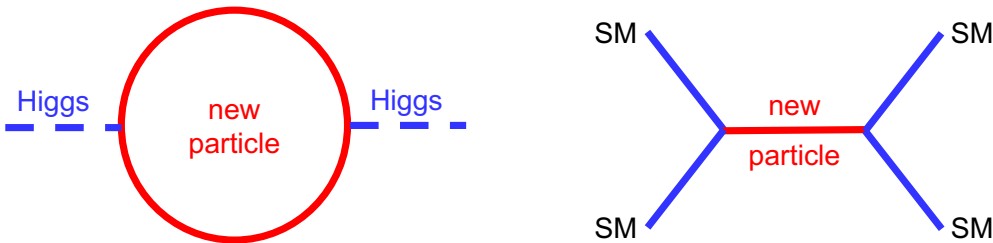

Figure 7: In many BSM models designed to explain the weak scale, new particles are introduced that contribute to the Higgs boson mass at 1-loop (left), but precision measurements from LEP severely constrain the contributions of these new particles to 4-point SM interactions (right).

The prototypical discrete parity is *R*-parity, proposed for SUSY long before the existence of LEP bounds [27]. However, the existence of LEP constraints implies that any new theory of the weak scale must confront this difficulty. The required discrete parity may be realized in many ways, depending on the new physics at the weak scale.

## 4 WIMPs in Supersymmetry

For the reasons mentioned above, WIMPs appear generically in many new physics models. The well-worn path is the following: (1) propose some new weak scale particles to solve some problem (the gauge hierarchy problem, the latest experimental anomaly, etc.), (2) realize that they also induce 4-point interactions shown in Fig. 7 and so, unfortunately, strain electroweak constraints and fits, (3) note that all these troubles can be ameliorated by imposing a discrete symmetry, (4) find that an ideal WIMP candidate emerges, and (5) declare victory (and promise a follow up paper exploring the implications for dark matter signals).

Rather than attempt an overview of all of the many examples, in this section we will dive in more detail into one of them by exploring the emergence of WIMPs in models with weak-scale SUSY.

### 4.1 Supersymmetry

The gauge hierarchy problem motivates supersymmetric extensions of the SM. In such models, every SM particle has a new, as-yet-undiscovered partner particle, which has the same quantum numbers and gauge interactions, but differs in spin by 1/2. The introduction of new particles with opposite spin-statistics from the known ones supplements the SM quantum corrections to the Higgs boson mass with opposite sign contributions, modifying the quantum corrections to the Higgs boson mass to be

$$\Delta m_h^2 \sim \frac{\lambda^2}{16\pi^2} \int^{\Lambda} \frac{d^4 p}{p^2}\bigg|_{\text{SM}} - \frac{\lambda^2}{16\pi^2} \int^{\Lambda} \frac{d^4 p}{p^2}\bigg|_{\text{SUSY}} \sim \frac{\lambda^2}{16\pi^2} \left(m_{\text{SUSY}}^2 - m_{\text{SM}}^2\right) \ln \frac{\Lambda}{m_{\text{SUSY}}}, \quad (7)$$

where $m_{\text{SM}}$ and $m_{\text{SUSY}}$ are the masses of the SM particles and their superpartners. For $m_{\text{SUSY}} \sim m_{\text{weak}}$, this is at most an $\mathcal{O}(1)$ correction, even for $\Lambda \sim M_{\text{Pl}}$. This by itself stabilizes, but does not solve, the gauge hierarchy problem; one must also understand why $m_{\text{SUSY}} \sim m_{\text{weak}} \ll M_{\text{Pl}}$. There are, however, a number of ways to generate such a hierarchy; for a review, see Ref. [28]. Given such a mechanism, quantum effects do not destroy the hierarchy, and the gauge hierarchy problem may be considered truly solved.

New Particle States

Stable

Standard Model
Particles

Figure 8: In BSM models with a conserved discrete parity under which new particles are odd and SM particles are even, heavy new particles may decay to the lightest new particle state, but the lightest new particle will be stable, as required for dark matter.

In more detail, the new particles predicted in SUSY include the
- Spin 3/2 gravitino
- Spin 1/2 gauginos: Bino, Winos, and gluinos
- Spin 1/2 Higgsinos
- Spin 0 squarks
- Spin 0 sleptons.

This represents a doubling of the number of particles in the SM. In fact, it is a bit more than a doubling. First there is the gravitino. But also the introduction of a new fermion, the Higgsino partner of the SM Higgs boson, introduces anomalies. To cancel these, another Higgs doublet is added, along with its superpartner Higgsinos. The resulting theory, the minimal supersymmetric standard model (MSSM), is, then, a supersymmetric extension of a two-Higgs-doublet extension of the SM.

In this list of new particles, only a few are both uncolored and electrically neutral and so potentially good dark matter candidates. These are

$$\text{Spin 3/2 Fermion:} \quad \text{Gravitino } \tilde{G} \tag{8}$$

$$\text{Spin 1/2 Fermions:} \quad \tilde{B}, \tilde{W}, \tilde{H}_u, \tilde{H}_d \to \text{Neutralinos } \chi_1, \chi_2, \chi_3, \chi_4 \tag{9}$$

$$\text{Spin 0 Scalars:} \quad \text{Sneutrinos } \tilde{\nu}_e, \tilde{\nu}_\mu, \tilde{\nu}_\tau . \tag{10}$$

As indicated, the neutral spin 1/2 fermions mix to form four mass eigenstates, the neutralinos. As we will see, the lightest of these, $\chi \equiv \chi_1$, is an excellent WIMP dark matter candidate [29, 30]. The sneutrinos are typically *not* good dark matter candidates, as both their annihilation and scattering cross sections are large, and so they are under-abundant or excluded by null results from direct detection experiments for all masses near $m_{\text{weak}}$ [31, 32].[2] The gravitino is not a WIMP, but it is a viable and fascinating dark matter candidate [33], as will be discussed in Sec. 6.3.

---

[2]Note, however, that right-handed sneutrinos may be good dark matter candidates [32].

## 4.2 Stability and LSPs

Not surprisingly, the introduction of so many particles has many implications. One of the first is a problem: the squarks mediate proton decay through renormalizable, dimension-4, interactions. These break both baryon and lepton number, and they mediate the decay $p \to \pi^0 e^+$ and others at unacceptably large rates. To forbid this, one introduces $R$-parity conversation, where $R_p = (-1)^{3(B-L)+2S}$ [27]. All SM particles have $R_p = 1$, and all superpartners have $R_p = -1$. Requiring that $R_p$ be conserved removes all interactions with an odd number of superpartners, including all those that mediate dimension-4 proton decay. And, of course, as anticipated in Sec. 3.3, it has the nice side effect of implying that the lightest supersymmetric particle (LSP) is stable, and a potential dark matter candidate.

What is, then, the LSP? To answer this question, we must discuss the importance of renormalization group (RG) evolution in SUSY. RG evolution has, of course, a central role in much of physics. In particle physics, we know that gauge couplings have different values depending on the scale at which they are probed. One can think of this as the effect of putting a charge in a dielectric, where in quantum field theory, even the vacuum is a dielectric. The most famous example of RG evolution in gauge couplings is asymptotic freedom, where the strong coupling evolves to lower values at higher energies, as has been verified to high accuracy in numerous experiments.

In SUSY, RG equations (RGEs) play an extremely important role. The leading example is in coupling constant unification. As is well known, the SM particles, with their rather strange gauge quantum numbers, fit beautifully into multiplets of grand unified theories (GUTs), such as SU(5) or SO(10). If grand unification is realized in nature, then the gauge couplings of the SM should also unify at some scale. This does not happen in the SM without additional particle content. But in the MSSM, the coupling constants unify at the grand unified scale $m_{\text{GUT}} \sim 10^{16}$ GeV [34]; see Fig. 9. This unification is far from trivial. Not only must three precisely-measured couplings unify, they must do so at a value that is in the perturbative regime ($\alpha_{\text{GUT}} \lesssim 1$) for the simple calculation to make sense, and they must do at a scale that is both below the Planck mass ($m_{\text{GUT}} \lesssim M_{\text{Pl}}$) and high enough that undesirable effects from GUT-scale physics (for example, dimension-5 and dimension-6 proton decay) do not violate experimental bounds (roughly, $m_{\text{GUT}} \gtrsim 10^{16}$ GeV).

Although coupling constant unification in supersymmetric GUT models is by far the most well-known feature of RGEs in SUSY, it is not the only one. All mass and coupling parameters RG evolve in SUSY; see Fig. 9. For many initial conditions at the GUT scale, the only mass parameter that runs negative is the mass$^2$ parameter of one of the Higgs multiplets, which breaks SU(2). This provides an explanation of why SU(2) is broken, but not SU(3) or U(1). In addition, the top quark Yukawa coupling has a quasi-fixed point, and so, for a large range of initial values at the GUT scale, it runs to $\lambda_t \simeq 1$ at the weak scale, providing an explanation of the top quark mass $m_t \simeq 173$ GeV.

Given all these nice features of RGEs in SUSY, what are the implications for the nature of the LSP? In the evolution of RG parameters, gauge couplings push masses to larger values and Yukawa couplings push masses to lower values. The expectation, then, is that the lightest superpartner is either the Bino among the gauginos, or the stau among all the squarks and sleptons. The electrically-charged stau is, of course, not a good WIMP candidate, but the neutral Bino is, and its emergence from the zoo of superpartners as a favored LSP candidate is a strong motivation to consider its implications for cosmology in great detail.

## 4.3 Neutralino Freezeout

If the Bino is WIMP dark matter, we can determine its thermal relic density by calculating its thermally averaged annihilation cross section $\langle \sigma_A v \rangle$ in any well-defined SUSY model. The

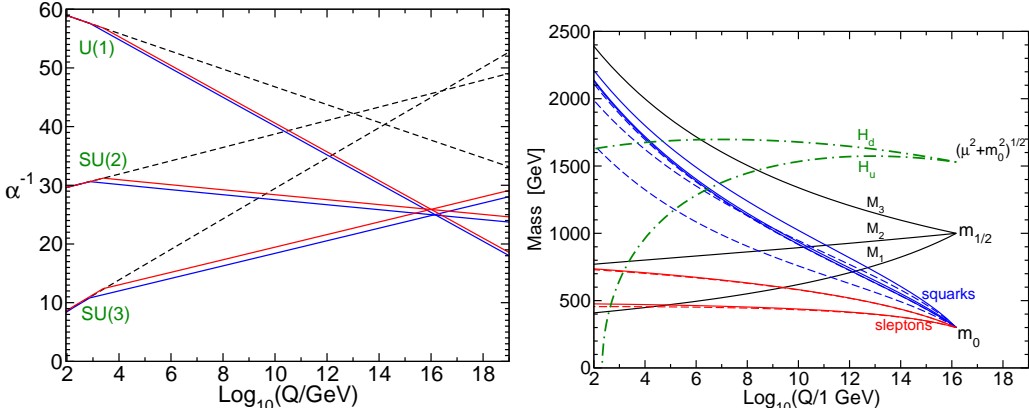

Figure 9: RG evolution in the MSSM. Left: The RG evolution of gauge couplings. In the SM (dashed), they do not unify, but in the MSSM (solid), they do unify at $Q \sim 10^{16}$ GeV. Right: The RG evolution of mass parameters in the MSSM. The mass$^2$ parameter for $H_u$ RG evolves to negative values, breaking SU(2). The Bino mass parameter $M_1$ often evolves to be the lowest of the remaining mass parameters, implying the Bino is the lightest SUSY particle and a natural dark matter candidate. From Ref. [35].

payoff of such a research program is clear:

- The regions of parameter space that give too much dark matter are excluded.
- The regions of parameter space that give too little are allowed, but Binos aren't all the dark matter.
- The regions of parameter space that give just the right amount of dark matter are cosmologically preferred and deserve special attention as one designs new experiments to search for dark matter, find SUSY at colliders, etc.

There are, unfortunately, a large number of processes through which neutralinos can annihilate; see Fig. 10. To bring order to this chaos, it is useful to begin by considering the pure Bino limit. The main annihilation diagrams may be divided into two classes:

- Annihilation to weak gauge bosons mediated by a $t$-channel charged Higgsinos and Winos. For pure Bino dark matter, these diagrams vanish. This follows from SUSY and the absence of 3-gauge boson vertices involving the hypercharge gauge boson.
- Annihilation to fermions mediated by $t$-channel sfermions. This reaction has an interesting structure. Neutralinos are Majorana fermions. If the initial state neutralinos are in an $S$-wave state, the Pauli exclusion principle implies that the initial state is $CP$-odd, with total spin $S = 0$ and total angular momentum $J = 0$. If the neutralinos are gauginos, the vertices preserve chirality, and so the final state $f\bar{f}$ has spin $S = 1$. This is compatible with $J = 0$ only with a mass insertion on the fermion line. This process is therefore either $P$-wave-suppressed (by a factor $v^2 \sim 0.1$) or chirality-suppressed (by a factor $m_f/m_W$).

The conclusion, then, is that for pure Binos, annihilation is typically suppressed, and the thermal relic density is therefore too large.

## 4.4 Cosmologically-Preferred Supersymmetry

There are, however, a number of interesting ways to enhance the annihilation cross section, and it is instructive to consider a few of these in the context of a well-defined SUSY model. A



Figure 10: Bino annihilation processes in the MSSM. From Ref. [12].

general supersymmetric extension of the SM contains many unknown parameters. To make progress, it is typical to consider specific models in which simplifying assumptions unify many parameters, and then study to what extent the conclusions may be generalized.

A simple example that is widely studied is minimal supergravity, sometimes called the constrained MSSM, which is minimal in the sense that it includes the minimum number of particles and includes a large number of assumptions that drastically reduces the number of independent model parameters. Minimal supergravity is defined by five parameters:

$$m_0, M_{1/2}, A_0, \tan\beta, \text{sign}(\mu). \tag{11}$$

The most important parameters are the universal scalar mass $m_0$ and the universal gaugino mass $M_{1/2}$, both defined at the scale of grand unified theories $m_{\text{GUT}} \simeq 2 \times 10^{16}$ GeV. The assumption of a universal gaugino mass and the choice of $m_{\text{GUT}}$ are supported by the fact that the three SM gauge couplings unify at $m_{\text{GUT}}$ in supersymmetric theories, as shown in Fig. 9. The assumption of scalar mass unification is much more *ad hoc*, but it does imply highly degenerate squarks and sleptons, which typically satisfies constraints on low-energy flavor- and CP-violation. Finally, the parameter $A_0$ governs the strength of cubic scalar particle interactions, and $\tan\beta$ and $\text{sign}(\mu)$ are parameters that enter the Higgs boson potential. For all but their most extreme values, these last three parameters have much less impact on collider and dark matter phenomenology than $m_0$ and $M_{1/2}$.

In the context of minimal supergravity, the thermal relic density is given in the $(m_0, M_{1/2})$ plane for fixed values of $A_0$, $\tan\beta$, and $\text{sign}(\mu)$ in Fig. 11. In the particular slice of parameter space shown, the Higgs boson mass is typically lower than the measured value $m_h \simeq 125$ GeV, but Fig. 11 will serve well to illustrate some qualitative features.

We see that current bounds on $\Omega_{\text{DM}}$ are highly constraining, essentially reducing the cosmologically favored parameter space by one dimension. For much of the region with $m_0, M_{1/2} \lesssim$ TeV, $\Omega_\chi$ is too large. This is because the lightest neutralino is Bino-like in much of the parameter space, and Bino-like annihilation is typically suppressed, as noted above. There are, however, regions with the correct thermal relic density:

- The "bulk region," in which the annihilation rate is boosted by light neutralinos and sleptons, with masses $m \lesssim 100$ GeV.

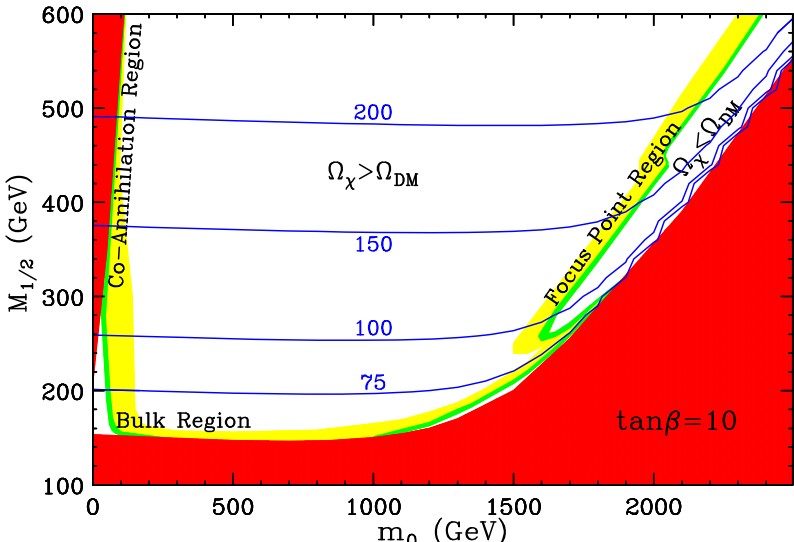

Figure 11: Regions of minimal supergravity $(m_0, M_{1/2})$ parameter space for fixed $A_0 = 0$, $\tan\beta = 10$, and $\mu > 0$. The green (yellow) region is cosmologically favored with $0.20 < \Omega_\chi < 0.28$ ($0.2 < \Omega_\chi < 0.6$). The names of cosmologically-favored regions (focus point, bulk, and co-annihilation) are indicated, along with regions with too much and too little dark matter. The lower right red shaded region is excluded by collider bounds on chargino masses; the upper left red region is excluded by the presence of a stable charged particle. Contours are for neutralino dark matter mass $m_\chi$ in GeV. Adapted from Ref. [36].

- The "focus point region," in which the lightest neutralino has a significant Higgsino component, and the annihilation to gauge bosons is no longer suppressed.
- The "co-annihilation region," in which the desired neutralino relic density may be obtained by neutralinos that co-annihilate with other particles that are present in significant numbers when the LSP freezes out. Naively, the presence of other particles requires that they be mass degenerate with the neutralino to within the temperature at freezeout, $T_f \sim m/20$. In fact, the co-annihilation cross section may be so enhanced relative to the neutralino-neutralino annihilation that it may be important even with mass splittings much larger than $T_f$.

If one considers the full minimal supergravity parameter space, other points in the $(m_0, M_{1/2})$ plane are possible (see, *e.g.*, Ref. [37]); notably, at larger $\tan\beta$ there is another favored region, known as the funnel region, in which neutralino annihilation is enhanced by a resonance through the CP-odd Higgs boson. The annihilation of pure Bino dark matter may also be enhanced by, for example, significant left-right sfermion mixing [38].

## 4.5 Are WIMPs Dead?

In the last decade, the LHC has started probing the weak scale, setting more and more stringent bounds on new physics. In addition, direct and indirect detection searches have become more stringent. Given all these constraints, are WIMPs now excluded?

To answer this question, let's first consider the minimal supergravity theories that we have discussed so far. In particular, what is the status of the various cosmologically-preferred scenarios discussed in Sec. 4.4? The bulk region requires sleptons lighter than 100 GeV, and is now largely excluded (see, however, Ref. [39]). On the other hand, the focus point region,

with mixed Higgsino-Bino neutralinos with masses up to 1 TeV, remains viable; it is most stringently probed by direct detection constraints (see Sec. 5.1), which have excluded some, but not all, of the parameter space. Last, the co-annihilation region remains unchallenged, as it requires only neutralinos and staus with masses $m \lesssim 600$ GeV, a mass range still outside of the LHC's reach. Such scenarios may also be used to resolve the muon $g-2$ anomaly. We see, then, that there are WIMP scenarios that are not only still viable, but even have additional nice features, such as SUSY to address the gauge hierarchy problem, minimality of field content, gauge coupling unification, scalar mass universality, and the desired thermal relic density.

One can, of course, consider WIMPs without one or more of these extra features. For example, working within SUSY, one can relax the various unification assumptions of minimal supergravity; for some recent discussions, see Refs. [40–42]. Alternatively, one can relax the constraint of minimality and introduce additional fields, such as a 4th generation of matter [43, 44] or additional singlets. One can also relax the constraint of the thermal relic density and consider a non-standard cosmology that produces dark matter in a different way. And, needless to say, one can work outside the framework of SUSY and address the gauge hierarchy problem in another way, or simply disregard it altogether.

All of these directions open up many, many more possibilities for WIMPs, which remain viable and highly motivated candidates for dark matter. For those with some familiarity with the field, then, the idea that WIMPs are now excluded is clearly very far from reality—wouldn't it be great if we lived in a world with such powerful technologies and experiments that we could exclude WIMPs! Unfortunately, we don't, and to misquote Mark Twain, rumors of the WIMP's death have been greatly exaggerated.

Before turning to WIMP searches in the following section, it is worth noting that, although, from the perspective of particle physics searches, SUSY can always remain alive by simply moving to higher masses, cosmology provides upper bounds on masses. SUSY particles cannot simply be decoupled, because this decoupling suppresses dark matter annihilation in the early universe, leading to too much dark matter now. This is an essential synergy between particle physics and cosmology, which uses the almost infinite energy provided by the hot Big Bang to probe scenarios with very heavy particles that cannot be probed by human-made experiments.

# 5 WIMP Detection

WIMP dark matter has many potential implications for search experiments, as was illustrated above in Fig. 4. In the following subsections, we discuss WIMP direct detection, indirect detection, and collider searches in turn.

## 5.1 Direct Detection

WIMP dark matter may be detected by its scattering off normal matter through processes $X$ SM $\rightarrow X$ SM. Given a typical WIMP mass of $m_X \sim 100$ GeV and WIMP velocity $v \sim 10^{-3}$, the deposited recoil energy is at most $\sim 100$ keV, requiring highly-sensitive, low-background detectors placed deep underground. Such detectors are insensitive to very strongly-interacting dark matter, which would be stopped in the atmosphere or earth and would be undetectable underground. However, very strongly-interacting dark matter would be seen by rocket and other space-borne experiments or would settle to the core of the Earth, leading to other fascinating and bizarre implications. Taken together, a diverse quilt of constraints now excludes large scattering cross sections for a wide range of dark matter masses (see Refs. [45, 46] and references therein), and we may concentrate on the weak cross section frontier probed by underground detectors.

For WIMP masses of around 100 GeV, the most stringent bounds are from searches for scattering off nuclei. (Scattering off electrons is particularly effective for light dark matter with masses at the GeV scale and below, and is now also being actively pursued [2,47].) Dark matter scattering off nuclei is induced by dark matter-quark interactions. For WIMPs such as neutralinos, the leading interactions are

$$\mathcal{L} = \sum_{q=u,d,s,c,b,t} \left( \alpha_q^{\text{SD}} \bar{X}\gamma^\mu\gamma^5 X \bar{q}\gamma_\mu\gamma^5 q + \alpha_q^{\text{SI}} \bar{X} X \bar{q} q \right) . \tag{12}$$

Given dark matter velocities now of $v \sim 10^{-3}$, we may consider these interactions in the non-relativistic limit. In this limit, the first terms reduce to spin-dependent couplings $S_X \cdot S_q$, and the second reduce to spin-independent couplings [48].

We will focus here on the spin-independent couplings. Experiments measure the dark matter-nucleus cross sections

$$\sigma_{\text{SI}} = \frac{4}{\pi}\mu_N^2 \sum_q \alpha_q^{\text{SI2}} \left[ Z\frac{m_p}{m_q}f_{T_q}^p + (A-Z)\frac{m_n}{m_q}f_{T_q}^n \right]^2 , \tag{13}$$

where

$$\mu_N = \frac{m_X m_N}{m_X + m_N} \tag{14}$$

is the reduced mass, and

$$f_{T_q}^p = \frac{\langle p|m_q\bar{q}q|p\rangle}{m_p} \tag{15}$$

is the fraction of the proton's mass carried by quark $q$, with a similar formula for neutrons. This may be parametrized by

$$\sigma_A = \frac{\mu_A^2}{M_*^4}\left[ f_p Z + f_n(A-Z) \right]^2 , \tag{16}$$

where $f_{p,n}$ are the nucleon level couplings, and $Z$ and $A-Z$ are the number of protons and neutrons in the nucleus, respectively. Note that at the typical energies of WIMP scattering, the dark matter sees the whole nucleus and does not resolve individual nucleons. Results are typically reported assuming $f_p = f_n$, so $\sigma_A \propto A^2$, and scaled to a single nucleon. With this assumption, scatterings off large nuclei are greatly enhanced. Note, however, that $f_p$ and $f_n$ are not necessarily equal, as we will discuss in Sec. 6.2, and all comparisons of scattering off different target nuclei are subject to this important caveat.

The event rate observed in a detector is, of course, also dependent on experimental and astrophysical details. For spin-independent detection, the rate is $R = \sigma_A I_A$, where

$$I_A = N_T\, n_X \int dE_R \int_{v_{\min}}^{v_{\text{esc}}} d^3v f(v)\frac{m_A}{2v\mu_A^2}F_A^2(E_R) , \tag{17}$$

where $N_T$ is the number of target nuclei, $n_X$ is the local dark matter number density, $E_R$ is the recoil energy, $f(v)$ is the local dark matter velocity distribution, with $v_{\text{esc}}$ the halo escape velocity, and $F_A$ is the nuclear physics form factor.

The field of direct detection is extremely active, with sensitivities increasing by an order of magnitude every few years over the last few decades. The current state of affairs is summarized in Fig. 12 for spin-independent searches. At present, the leading bounds are from one- to multi-tonne-scale liquid noble gas detectors, including XENON1T [49], PandaX-4T [50], and LZ [51]. For dark matter masses $\sim 20-100$ GeV, the upper bound on the dark matter-nucleon cross section, assuming $f_p = f_n$, is at the $10^{-47}$ cm$^2$ level.

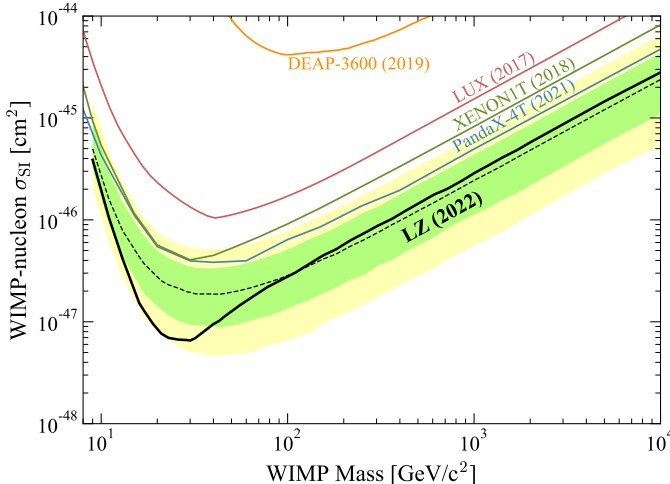

Figure 12: Upper bounds on spin-independent WIMP-nucleon cross sections [51]. The solid contours show the 90% confidence limits from the experiments indicated. The green and yellow bands are the $1\sigma$ and $2\sigma$ sensitivity bands for LZ, and the dashed line shows the median of the LZ sensitivity projection.

How significant is this progress? The current bounds are probing the heart of WIMP theory parameter space, with many otherwise successful WIMP theories being excluded by direct detection. At the same time, there are WIMP theories with almost arbitrarily small cross sections. Another way to answer the question is to consider how close we are coming to the ultimate limits of direct detection experiments. Although here, too, one can imagine almost arbitrarily sensitive experiments, an irreducible background to non-directional direct detection experiments is provided by the flux of solar, atmospheric, and diffuse supernovae neutrinos. These provide a "neutrino fog," beyond which it will be much more difficult to probe [52, 53]. This limit of background-free, non-directional direct detection searches (and also the metric prefix system!) will be reached when spin-independent cross sections reach this background neutrino signal at cross sections of of $\sim 1$ yb ($10^{-3}$ zb, $10^{-12}$ pb, or $10^{-48}$ cm$^2$), which will be probed by $\sim 10$-tonne experiments in the coming years.

In addition to the limits described above, the DAMA experiment continues to find a signal in annual modulation [54] with period and maximum at the expected values [55]. The annual modulation signal arises from the motion of the Earth around the Sun, which results in greater scattering rates at certain times of the year. The required mass is rather low for WIMPs, $m_X \sim 1 - 10$ GeV, and the required cross section is very high, $\sigma_{\rm SI} \sim 10^{-41} - 10^{-39}$ cm$^2$. Such parameters are excluded by other experiments in most, if not all, model frameworks, and the DAMA results are now also being tested by other experiments that use the same NaI target material. As we will see in Sec. 6, however, the DAMA signal, whatever its ultimate fate, has been a fantastic driver of new theoretical ideas, motivating, for example, the ideas of inelastic dark matter [56] and isospin-violating dark matter [57, 58], which have general applicability well beyond simply trying to reconcile the DAMA signal with other null results.

## 5.2 Indirect Detection

After freezeout, dark matter pair annihilation becomes greatly suppressed. However, even if its impact on the dark matter relic density is negligible, dark matter annihilation continues and may be observable. Dark matter may therefore be detected indirectly: dark matter pair-annihilates somewhere, producing something, which is detected somehow. There are many

indirect detection methods being pursued [59, 60]. Their relative sensitivities are highly dependent on what WIMP candidate is being considered, and the systematic uncertainties and difficulties in determining backgrounds also vary greatly from one method to another.

Assuming dark matter annihilation is $S$-wave, and so the thermally-averaged cross section is approximately velocity-independent, $\langle \sigma_A v \rangle \sim \sigma_0$, the correct relic density is achieved by thermal freezeout for $\sigma_0 \approx 2$ to $3 \times 10^{-26}$ cm$^3$/s. This provides an important target for indirect searches. Of course, if the annihilation is $P$-wave, the correspondence between the annihilate cross section at freezeout and now is broken. Nevertheless, relative to direct detection, indirect rates typically have smaller particle physics uncertainties (but larger astrophysical uncertainties), since annihilation determines both the relic density and the rate.

Among the most interesting indirect searches are those for photons produced in WIMP annihilation. There are two kinds of such signals. Line signals may be produced by $XX \to \gamma\gamma, \gamma Z$. Since WIMPs do not couple to photons directly, these processes are loop-suppressed, but, given that dark matter is highly non-relativistic now, they lead to a very distinctive signal of monoenergetic photons. Alternatively, continuum signals may be produced by $XX \to f\bar{f}$, where a photon is radiated from one of the fermion final states. The continuum signal is less distinctive, but the rates are larger than the line signal.

Since gamma ray photons have such high energies, they are typically not deflected and thus point back to their source, providing a powerful diagnostic. Possible targets for gamma ray searches are the center of the galaxy, where signal rates are high, but backgrounds are also high and potentially hard to estimate; and dwarf galaxies, where signal rates are lower, but backgrounds are also expected to be low. A possible excess from a continuum signal from the galactic center has generated interest since its original observation [61].

The leading searches for gamma rays from WIMP annihilation are space-based experiments, such as Fermi-LAT [62, 63] and AMS [64], and ground-based atmospheric Cherenkov telescopes, such as the Cherenkov Telescope Array (CTA) [65]. Based on null results for searches for the continuum signal, Fermi-LAT has already excluded light WIMPs with the target annihilation cross section. For example, WIMPs that decay through $XX \to b\bar{b}$ are excluded by galactic center searches for WIMPs up to tens of GeV. In the future, CTA is expected to be sensitive to WIMPs with the target annihilation cross section and masses from 100 GeV to 10 TeV.

Searches for neutrinos are unique among indirect searches in that they are, given certain assumptions, probes of scattering cross sections, not annihilation cross sections, and so compete directly with direct detection searches. The idea behind neutrino searches is the following: when WIMPs pass through the Sun or the Earth, they may scatter and be slowed below escape velocity. Once captured, they then settle to the center, where their densities and annihilation rates are greatly enhanced. Although most of their annihilation products are immediately absorbed, neutrinos are not. Some of the resulting neutrinos then travel to the surface of the Earth, where they may convert to charged leptons through $\nu q \to \ell q'$, and the charged leptons may be detected.

The neutrino flux depends on the WIMP density, which is determined by the competing processes of capture and annihilation. If $N$ is the number of WIMPs captured in the Earth or Sun, $\dot{N} = C - AN^2$, where $C$ is the capture rate and $A$ is the total annihilation cross section times relative velocity per volume. The present WIMP annihilation rate is, then, $\Gamma_A \equiv AN^2/2 = C \tanh^2(\sqrt{CA} t_\odot)/2$, where $t_\odot \simeq 4.5$ Gyr is the age of the solar system. For most WIMP models, a very large collecting body such as the Sun has reached equilibrium, and so $\Gamma_A \approx C/2$. The annihilation rate alone does not completely determine the differential neutrino flux — one must also make assumptions about how the neutrinos are produced. However, if one assumes, say, that WIMPs annihilate to $b\bar{b}$ or $W^+W^-$, which then decay to neutrinos, as is true in many neutralino models, the neutrino signal is completely determined

by the capture rate $C$, that is, the scattering cross section.

Under fairly general conditions, then, neutrino searches are directly comparable to direct detection. For example, the Super-Kamiokande [66] and IceCube [67] Collaborations have looked for excesses of neutrinos from the Sun with energies in the range $10\,\text{GeV} \lesssim E_\nu \lesssim 1\,\text{TeV}$. Given the assumptions specified above, their null results provide leading bounds on spin-dependent scattering cross sections. These experiments are just beginning to probe relevant regions of supersymmetric and UED parameter space. Neutrino searches are also sensitive to spin-independent cross sections, but for typical WIMP masses, they are not competitive with direct searches.

As a last example, dark matter may annihilate in the galactic halo to anti-matter, which can be detected by, for example, the space-based detectors Fermi-LAT [62, 63] and AMS [64]. In contrast to gamma ray photons and neutrinos, anti-matter, such as positrons, anti-protons, and anti-deuterons, do not travel in straight lines, but rather bump around in the local halo before arriving in our detectors. These signals are therefore less distinctive, and they may not be easy to disentangle from astrophysical backgrounds, such as pulsars. At present, there are, however, anomalies at AMS, notably a few anti-$^3$He and anti-$^4$He nuclei that have been reported [68–70]. The expected rates from astrophysics for these exotic anti-nuclei is typically far below what is seen, and if verified, they may be regarded as smoking gun signals of dark matter.

## 5.3 Collider Searches

If WIMPs are the dark matter, what can colliders tell us? Given the energy of the LHC and the requirement that WIMPs have mass $\sim m_{\text{weak}}$ and interact through the weak force, WIMPs will almost certainly be produced at the LHC. Unfortunately, direct WIMP production of $XX$ pairs is invisible, and so one must look for signatures of WIMPs produced in conjunction with other particles.

In SUSY, the LHC will typically produce pairs of squarks and gluinos. These will then decay through some cascade chain, eventually ending up in neutralino WIMPs, which escape the detector. Their existence is registered through the signature of missing energy and momentum, a signal that is a staple of searches for physics beyond the SM. Analyses of this type may be carried out with fully-defined supersymmetric models, or with pared down, so-called "simplified models", in which dark matter and just a few other particles are introduced, with just a few defining parameters [71, 72].

Alternatively, one may produce WIMPs directly, but in association with something else that can be seen. Such analyses are typically carried out with an effective theory. For example, one can assume an effective theory interaction $q\bar{q}XX$, and look for production of $XX$ in association with a gluon or photon radiated from an initial state quark, leading to a mono-jet or mono-photon signal. Systematic analyses for various types of WIMP dark matter and all possible 4-point effective interactions have been carried out, leading to LHC bounds on all such effective operators [73–75]. The effective theory approach allows comparisons between direct detection, indirect detection, and collider searches with various signatures, but it requires that the effective theory is valid, that is, that the mediator particle that induces these effective operators be heavy relative to the available energies. This is not always true at the LHC [76].

Last, it is important to note that, even if a new particle is observed at a collider through the missing energy or momentum signature, this would be far from compelling evidence for dark matter. The observation of missing particles only implies that a particle was produced that was stable enough to exit the detector, typically implying a lifetime $\tau \gtrsim 10^{-7}$ s, 24 orders of magnitude from the criterion $\tau \gtrsim 10^{17}$ s required for dark matter. If a signal is seen at a collider, corroborating evidence from, say, direct detection of a particle with a similar mass would be required to establish that the collider signal had cosmological relevance.

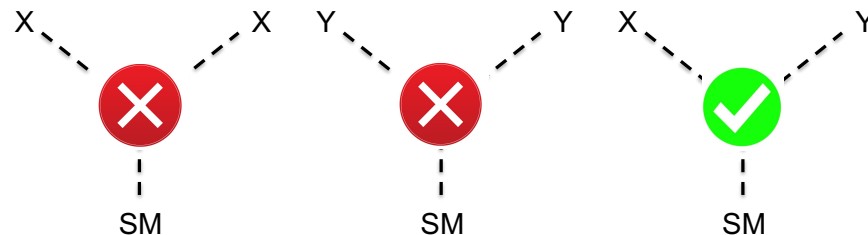

Figure 13: In inelastic dark matter, two highly-degenerate states $X$ and $Y$ couple only off-diagonally to the SM.

## 6 Variations

As described above, WIMPs have been extremely fertile ground for encouraging new ideas for how to search for dark matter particles. At the same time, the WIMP paradigm has also generated a great deal of theoretical work, leading to particle physics models that preserve some of the WIMP's main features, for example, the WIMP miracle, but have very different implications for particle physics experiments and astrophysical observations. In this section, we give a few example of these variations on the WIMP paradigm.

### 6.1 Inelastic WIMPs

As noted above, the DAMA signal has been a fantastic generator of new ideas for particle dark matter. Perhaps the most prominent is the idea of inelastic dark matter [56]. This scenario grew out of considerations of another SUSY WIMP candidate, the (messenger) sneutrino [77, 78], a complex scalar, which may be split into two highly-degenerate, real scalar states.

In the inelastic dark matter scenario, one considers two highly-degenerate particles with WIMP-like masses and interactions, $X$ and $Y$, which couple only off-diagonally to the SM; see Fig. 13. For masses $m_X, m_Y \sim 100$ GeV and mass splitting $\Delta \equiv m_Y - m_X \sim$ MeV, these particles freeze out in the early universe as usual, since the mass splitting is negligible relative to the temperature at freezeout. After freezeout, as the universe continues to cool, all of the $Y$ particles decay to $X$ particles, producing $X$ particles that may naturally have the correct relic density through the WIMP miracle.

Fast forwarding to today, however, since there are no $Y$ particles in the universe and no $X - X - $ SM couplings, indirect detection signals are suppressed. Additionally, the absence of $X - X - $ SM couplings eliminates $Xq \to Xq$ scattering for direct detection, and $Xq \to Yq$ scattering is also suppressed, since local dark matter, with velocities $v \sim 10^{-3}$ and kinetic energies $\sim 100$ keV, does not have enough energy to up-scatter to produce $Y$ particles. For tuned values of $\Delta \sim 100$ keV, one can suppress scattering in germanium and preserve scattering off iodine, which was used at one time to reconcile the signal seen by DAMA with null results from CDMS [56]. But more generally, inelastic dark matter preserves the virtue of the WIMP miracle, while nullifying indirect and direct searches, opening up new parameter space to novel searches, for example, at accelerators and colliders.

### 6.2 Isospin-Violating WIMPs

The standard presentation of direct detection experimental results for spin-independent scattering is in the $(m_X, \sigma_p)$ plane, where $m_X$ is the mass of the dark matter particle $X$, and $\sigma_p$ is the $X$-proton scattering cross section. However, direct detection experiments do not directly constrain $\sigma_p$. Rather, they bound scattering cross sections off of nuclei. As noted above, results for nuclei are then interpreted as bounds on $\sigma_p$ by assuming that the couplings of dark

matter to protons and neutrons are identical, *i.e.*, that the dark matter's couplings are isospin-invariant.

This assumption is valid if the interaction between dark matter and quarks is mediated by a Higgs boson, as in the case of neutralinos with heavy squarks. In general, however, it is not theoretically well-motivated: the assumption is violated by many dark matter candidates, including neutralinos with light squarks, dark matter with $Z$-mediated interactions with the SM, dark matter charged under a hidden U(1) gauge group with a small kinetic mixing with hypercharge, and dark matter coupled through new scalar or fermionic mediators with arbitrary flavor structure. See, for example, Ref. [79].

Isospin-violating dark matter (IVDM) [57, 58] provides a simple framework that accommodates all these possibilities by including a single new parameter, the neutron-to-proton coupling ratio $f_n/f_p$. One might have expected an overarching framework to need many more parameters. However, for spin-independent scattering with the typical energies of weakly-interacting massive particle (WIMP) collisions, the dark matter does not probe the internal structure of nucleons. Nucleons are therefore the correct effective degrees of freedom for spin-independent WIMP scattering, and IVDM therefore captures all of the possible variations by letting the proton couplings differ from the neutron couplings.

Dark matter-nuclei scattering is largely coherent, which for isospin-invariant scenarios produces a well-known $A^2$ enhancement to the cross section, favoring scattering off heavier elements. But in the case of isospin violation, destructive interference can instead suppress the scattering cross section. Although direct detection experiments typically present results in terms of $\sigma_p$, the actual quantity reported is the *normalized-to-nucleon cross section* $\sigma_N^Z$, which is the dark matter-nucleon scattering cross section that is inferred from the data of a detector with a target with $Z$ protons, assuming isospin-invariant interactions. This quantity is related to $\sigma_p$ by the "degradation factor" [58]

$$D_p^Z \equiv \frac{\sigma_N^Z}{\sigma_p} = \frac{\sum_i \eta_i \mu_{A_i}^2 [Z + (f_n/f_p)(A_i - Z)]^2}{\sum_i \eta_i \mu_{A_i}^2 A_i^2}, \qquad (18)$$

where $\eta_i$ is the natural abundance of the $i^{\text{th}}$ isotope, $\mu_{A_i} = m_X m_{A_i}/(m_X + m_{A_i})$ is the reduced mass of the dark matter-nucleus system, and $f_n$ and $f_p$ are the couplings of dark matter to neutrons and protons, respectively, as discussed above in Sec. 5.1. For isospin-invariant interactions, $f_n = f_p$, and $\sigma_N^Z = \sigma_p$.

Absent any prejudice, $f_n/f_p$ is a free parameter that must be constrained by data, no different than the mass and cross section. But we can identify some benchmark values of $f_n/f_p$ that are particularly noteworthy:

- $f_n/f_p = -13.3$ ("$Z$-mediated"): Valid for dark matter with $Z$-mediated interactions with the SM.
- $f_n/f_p = -0.82$ ("Argophobic"): For this value, the sensitivity of argon-based detectors is maximally degraded.
- $f_n/f_p = -0.70$ ("Xenophobic"): For this value, the sensitivity of xenon-based detectors is maximally degraded [58, 80].
- $f_n/f_p = 0$ ("Dark photon-mediated"): Valid for dark matter that interacts with the SM through kinetic mixing with the photon.
- $f_n/f_p = 1$ ("Isospin-invariant"): Valid for dark matter that interacts with the SM through Higgs exchange.

In Fig. 14 we plot the degradation factor $\sigma_N^Z/\sigma_p$ as a function of $f_n/f_p$ for many of the target materials commonly used in direct detection experiments. The full range of $f_n/f_p$ is shown in the left panel, and the xenophobic region near $f_n/f_p = -0.70$ is shown in the right panel. For

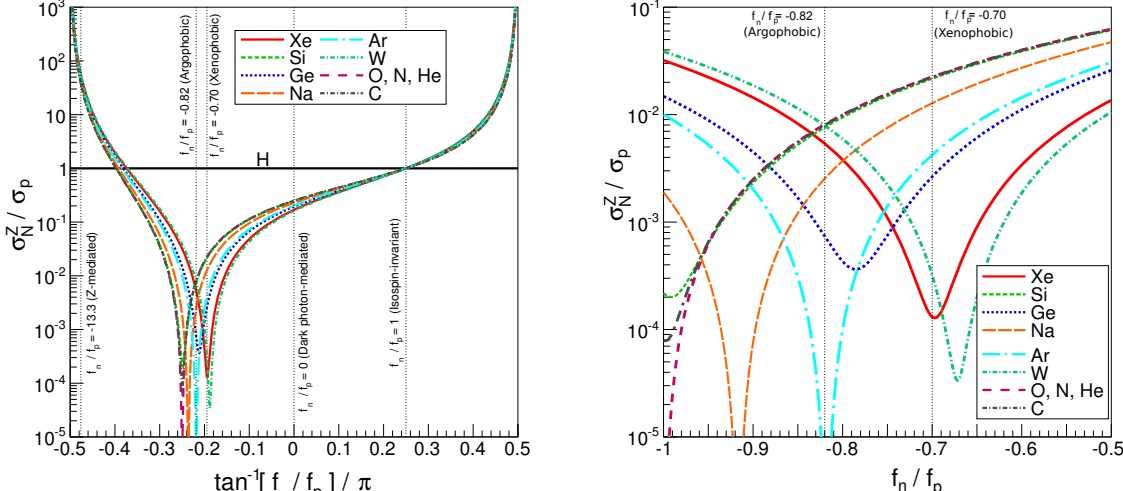

Figure 14: Ratio of $\sigma_Z^N$ to $\sigma_p$ for materials relevant to direct detection experiments [58]. Ratios are shown as a function of $f_n/f_p$ for the entire range of couplings (left) and the xenophobic region (right). We have made the mild assumption that the reduced masses $\mu_{A_i}$ are all equal for a given element and dark matter mass.

materials with only one isotope with significant abundance, such as oxygen, nitrogen, helium, sodium, and argon, it is possible to almost completely eliminate the detector's response with a particular choice of $f_n/f_p$. But for a material such as xenon, with many isotopes, it is not possible to cancel the response of all isotopes simultaneously. For materials such as carbon, silicon, germanium, xenon, and tungsten, the maximum factor by which their sensitivity to $\sigma_p$ may be degraded is within the range $10^{-5} - 10^{-3}$.

## 6.3 SuperWIMPs

The WIMP miracle might appear to require that dark matter have weak interactions if its relic density is naturally to be in the right range. This is not true, however. In this section, we discuss superWIMPs [81, 82], superweakly-interacting massive particles, which have the desired relic density, but have interactions that are much weaker than weak. The extremely weak interactions of superWIMPs are, in some respects, a nightmare for searches for dark matter. At the same time, however, superWIMP scenarios predict signals at colliders and in astrophysics that can be far more striking than in WIMP scenarios, making superWIMPs amenable to entirely different investigations.

In the superWIMP framework, dark matter is produced in late decays: WIMPs freeze out as usual in the early universe, but later decay to superWIMPs, which form the dark matter that exists today. Because superWIMPs are very weakly interacting, they have no impact on WIMP freezeout in the early universe, and the WIMPs decouple, as usual, with a thermal relic density $\Omega_{\text{WIMP}} \sim \Omega_{\text{DM}}$. Assuming that each WIMP decay produces one superWIMP, the relic density of superWIMPs is

$$\Omega_{\text{SWIMP}} = \frac{m_{\text{SWIMP}}}{m_{\text{WIMP}}} \Omega_{\text{WIMP}} . \tag{19}$$

SuperWIMPs therefore inherit their relic density from WIMPs, and for $m_{\text{SWIMP}} \sim m_{\text{WIMP}}$, the WIMP miracle also implies that superWIMPs are produced in the desired amount to be much or all of dark matter. The evolution of number densities is shown in Fig. 15. The WIMP decay may be very late by particle physics standards. For example, if the superWIMP interacts only gravitationally, as is true of many well-known candidates, the natural timescale for WIMPs decaying to superWIMP is $1/(G_N m_{\text{weak}}^3) \sim 10^3 - 10^7$ s.

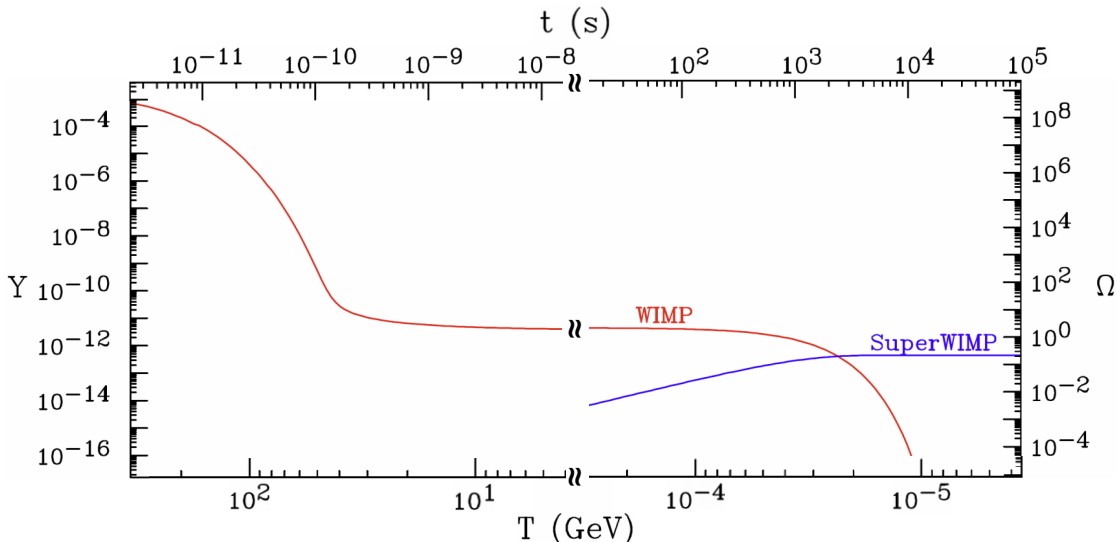

Figure 15: In superWIMP scenarios, WIMPs freeze out as usual, but then decay to superWIMPs, superweakly-interacting massive particles that are the dark matter. This figure shows the WIMP comoving number density $Y$ (left) and the superWIMP relic density (right) as functions of temperature $T$ (bottom) and time $t$ (top). The WIMP is a 1 TeV, $S$-wave annihilating particle with lifetime $10^3$ s, and the superWIMP has mass 100 GeV.

Because the WIMP is unstable and not the dark matter, it need not be neutral in this context: to preserve the naturalness of the superWIMP relic density, all that is required is $\Omega_{\text{WIMP}} \sim \Omega_{\text{DM}}$. In the case of SUSY, for example, the WIMP may be a neutralino, but it may also be a charged slepton. Even though charged sleptons interact with photons, on dimensional grounds, their annihilation cross sections are also necessarily governed by the weak scale, and so are $\sim g_{\text{weak}}^4/m_{\text{weak}}^2$, implying roughly the same relic densities as their neutral counterparts. Of course, whether the WIMP is charged or not determines the properties of the other particles produced in WIMP decay, which has strong consequences for observations, as we will see below.

The superWIMP scenario is realized in many particle physics models. The prototypical superWIMP is the gravitino $\tilde{G}$ [81–86]. Gravitinos are the spin 3/2 superpartners of gravitons, and they exist in all supersymmetric theories. The gravitino's mass is

$$m_{\tilde{G}} = \frac{F}{\sqrt{3}M_*}\,, \tag{20}$$

where $F$ is the SUSY-breaking scale squared and $M_* = (8\pi G_N)^{-1/2} \simeq 2.4 \times 10^{18}$ GeV is the reduced Planck mass. In the simplest supersymmetric models, where SUSY is transmitted to SM superpartners through gravitational interactions, the masses of SM superpartners are

$$\tilde{m} \sim \frac{F}{M_*}\,. \tag{21}$$

A solution to the gauge hierarchy problem requires $F \sim (10^{11}\text{ GeV})^2$, and so all superpartners and the gravitino have weak-scale masses. The precise ordering of masses depends on unknown, presumably $\mathcal{O}(1)$, constants in Eq. (21). There is no theoretical reason to expect the gravitino to be heavier or lighter than the lightest SM superpartner, and so in roughly "half" of the parameter space, the gravitino is the lightest supersymmetric particle (LSP). Its stability is

guaranteed by $R$-parity conservation, and since $m_{\tilde{G}} \sim \tilde{m}$, the gravitino relic density is naturally $\Omega_{\text{SWIMP}} \sim \Omega_{\text{DM}}$.

In gravitino superWIMP scenarios, the role of the decaying WIMP is played by the next-to-lightest supersymmetric particle (NLSP), a charged slepton, sneutrino, chargino, or neutralino. The gravitino couples SM particles to their superpartners through gravitino-sfermion-fermion interactions

$$L = -\frac{1}{\sqrt{2}M_*} \partial_\nu \tilde{f} \, \bar{f} \, \gamma^\mu \gamma^\nu \tilde{G}_\mu \,, \tag{22}$$

and gravitino-gaugino-gauge boson interactions

$$L = -\frac{i}{8M_*} \bar{\tilde{G}}_\mu [\gamma^\nu, \gamma^\rho] \gamma^\mu \tilde{V} F_{\nu\rho} \,. \tag{23}$$

The presence of $M_*$ in Eqs. (22) and (23) implies that gravitinos interact only gravitationally, a property dictated by the fact that they are the superpartners of gravitons. These interactions determine the NLSP decay lifetime. As an example, if the NLSP is a stau, a superpartner of the tau lepton, its lifetime is

$$\tau(\tilde{\tau} \to \tau \tilde{G}) = \frac{6}{G_N} \frac{m_{\tilde{G}}^2}{m_{\tilde{\tau}}^5} \left[1 - \frac{m_{\tilde{G}}^2}{m_{\tilde{\tau}}^2}\right]^{-4} \approx 3.7 \times 10^7 \text{ s} \left[\frac{100 \text{ GeV}}{m_{\tilde{\tau}} - m_{\tilde{G}}}\right]^4 \left[\frac{m_{\tilde{G}}}{100 \text{ GeV}}\right], \tag{24}$$

where the approximate expression holds for $m_{\tilde{G}}/m_{\tilde{\tau}} \approx 1$. We see that decay lifetimes of the order of hours to months are perfectly natural. At the same time, the lifetime is quite sensitive to the underlying parameters and may be much longer for degenerate $\tilde{\tau} - \tilde{G}$ pairs, or much shorter for light gravitinos.

In contrast to WIMPs, superWIMPs are produced with large velocities at late times. This has two, *a priori* independent, effects. First, the velocity dispersion reduces the phase space density, smoothing out cusps in dark matter halos. Second, such particles damp the linear power spectrum, reducing power on small scales [87–93]. As seen in Fig. 16, superWIMPs may suppress small scale structure as effectively as a 1 keV sterile neutrino, a famously warm dark matter candidate. Some superWIMP scenarios may therefore be differentiated from standard cold dark matter scenarios by their impact on small scale structure, and the late decays of WIMPs to superWIMPs may also produce signals in Big Bang nucleosynthesis and the cosmic microwave background [90, 91].

Particle colliders may also find evidence for superWIMP scenarios. If the decaying WIMP is charged, the superWIMP scenario predicts long-lived, charged particles at colliders. If they are stable in the timescales of seconds to months, one can collect these particles and study their decays. Several ideas have been proposed. One can catch the metastable charged particles in an auxiliary detector, such as a water tank, placed just outside the ATLAS or CMS detectors, and then transport the water to a quiet location to observe the eventual charged particle decay [94]. Alternatively, one can catch the charged particles in LHC detectors themselves, and look for decays, say, when the beams are off [95], or it has even been proposed to let the charged particles lodge themselves in the detector hall walls and, through precision measurements, determine their locations and dig them out of these walls [96]. The search for long-lived particles at colliders has in recent years received renewed attention, with superWIMP scenarios being just one of many interesting motivations [97, 98].

## 6.4 WIMPless Dark Matter

As discussed in Sec. 3.2, the thermal relic density of a stable particle with mass $m_X$ annihilating through interactions with couplings $g_X$ is

$$\Omega_X \sim \langle \sigma_A v \rangle^{-1} \sim \frac{m_X^2}{g_X^4} \,. \tag{25}$$

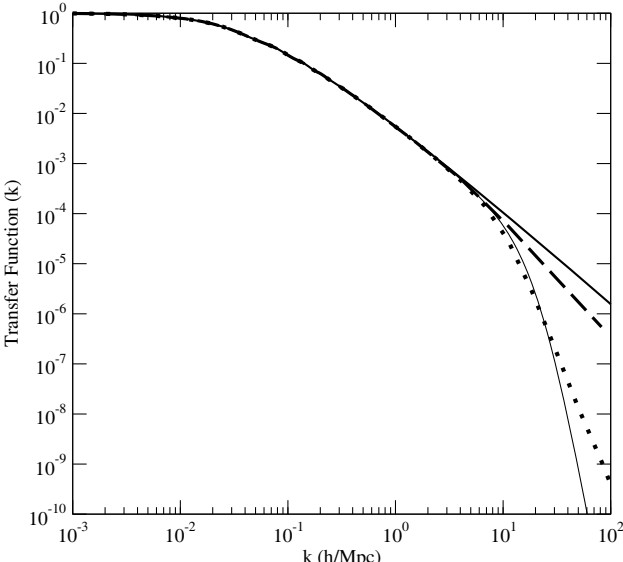

Figure 16: The power spectrum for scenarios in which dark matter is completely composed of WIMPs (solid), half WIMPs and half superWIMPs (dashed), and completely composed of superWIMPs (dotted). For comparison, the lower solid curve is for 1 keV sterile neutrino warm dark matter. From Ref. [90].

The WIMP miracle is the fact that, for $m_X \sim m_{\text{weak}}$ and $g_X \sim g_{\text{weak}} \simeq 0.65$, $\Omega_X$ is roughly $\Omega_{\text{DM}} \approx 0.23$.

Equation (25) makes clear, however, that the thermal relic density fixes only one combination of the dark matter's mass and coupling, and other combinations of $(m_X, g_X)$ can also give the correct $\Omega_X$. This can be seen in Fig. 3, where the parameter space with the correct relic density is not a point in the $(M, g)$ plane, but a line. In the SM, $g_X \sim g_{\text{weak}}$ is the only choice available, but in a general hidden sector, with its own matter content and gauge forces, other values of $(m_X, g_X)$ may be realized. Such models generalize the WIMP miracle to the "WIMPless miracle" [8]: dark matter that naturally has the correct relic density, but does not necessarily have a weak-scale mass or weak interactions.

The WIMPless miracle is naturally realized in particle physics frameworks that have several other motivations. A well-known example is supersymmetric models with gauge-mediated SUSY breaking (GMSB). These models necessarily have several sectors, as shown in Fig. 17. The SUSY-breaking sector includes the fields that break SUSY dynamically and the messenger particles that mediate this breaking to other sectors through gauge interactions. The MSSM sector includes the fields of supersymmetric extension of the SM. In addition, SUSY breaking may be mediated to one or more hidden sectors. The hidden sectors are not strictly necessary, but there is no reason to prevent them, and hidden sectors are ubiquitous in such models originating in string theory. GMSB models generically predict a Higgs boson lighter than the measured value, but this may be rectified by heavy superpartners [99] or extra field content [100].

The essential feature of GMSB models is that they elegantly suppress troublesome contributions to flavor-violating processes by introducing generation-independent squark and slepton masses of the form

$$m \sim \frac{g^2}{16\pi^2} \frac{F}{M_{\text{m}}} \,. \tag{26}$$

The generic feature is that superpartner masses are proportional to gauge couplings squared times the ratio $F/M_{\text{m}}$, which is a property of the SUSY-breaking sector. With analogous cou-

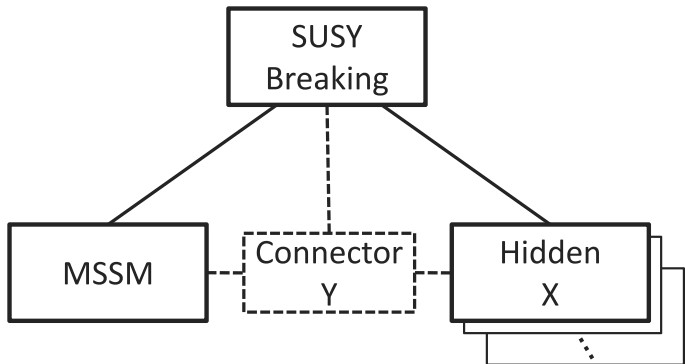

Figure 17: Sectors of supersymmetric models. SUSY breaking is mediated by gauge interactions to the MSSM and the hidden sector, which contains the dark matter particle $X$. An optional connector sector contains fields $Y$, charged under both MSSM and hidden sector gauge groups, which induce signals in direct and indirect searches and at colliders. There may also be other hidden sectors, leading to multi-component dark matter. From Ref. [8].

plings of the hidden sector fields to hidden messengers, the hidden sector superpartner masses are

$$m_X \sim \frac{g_X^2}{16\pi^2}\frac{F}{M_m}, \tag{27}$$

where $g_X$ is the relevant hidden sector gauge coupling. As a result,

$$\frac{m_X}{g_X^2} \sim \frac{m}{g^2} \sim \frac{F}{16\pi^2 M_m} ; \tag{28}$$

that is, $m_X/g_X^2$ is determined solely by the SUSY-breaking sector. As this is exactly the combination of parameters that determines the thermal relic density of Eq. (25), the hidden sector automatically includes a dark matter candidate that has the desired thermal relic density, irrespective of its mass. This has been verified numerically for a concrete hidden sector model [101, 102]; the results are shown in Fig. 18. This property relies on the relation $m_X \propto g_X^2$, which may also be found in other settings [103, 104].

As is evident from Fig. 18, WIMPless dark matter opens up the possibility of light dark matter with masses in the MeV to GeV range that still has the virtue of being produced through thermal freezeout with the correct relic density. A large number of experiments are now planned or underway to look for such light dark matter [2, 97].

WIMPless and other hidden sector models also naturally open the possibility of dark forces in the hidden sector. In the WIMPless scenarios just described, this possibility arises naturally if one attempts to understand why the hidden sector particle is stable. This is an important question; after all, in these GMSB models, all SM superpartners decay to the gravitino. In the hidden sector, an elegant way to stabilize the dark matter is through U(1) charge conservation. This possibility necessarily implies massless gauge bosons in the hidden sector. Alternatively, the hidden sector may have light, but not massless, force carriers. In all of these cases, the dynamics of the hidden sector may have many interesting astrophysical implications, naturally predicting self-interacting dark matter, which has numerous important astrophysical signatures and may even be indicated by observational data [105–107].

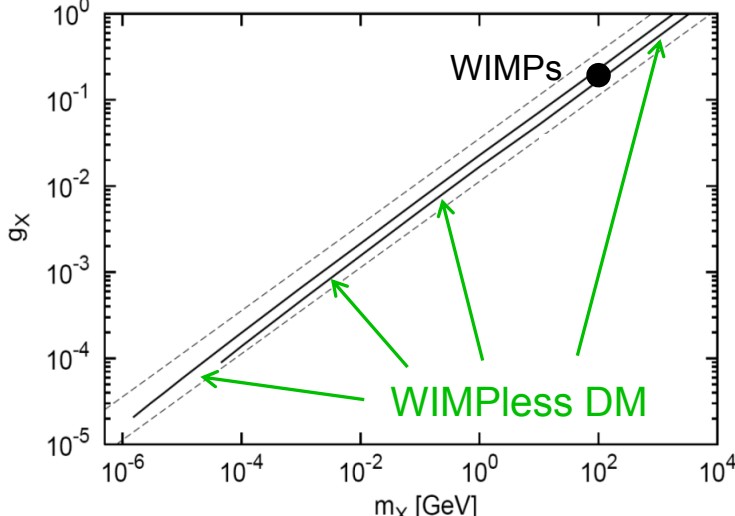

Figure 18: Contours of $\Omega_X h^2 = 0.11$ in the $(m_X, g_X)$ plane for hidden to observable reheating temperature ratios $T_{\mathrm{RH}}^h/T_{\mathrm{RH}} = 0.8$ (upper solid) and 0.3 (lower solid), where the hidden sector is a 1-generation flavor-free version of the MSSM. Also plotted are lines of $m_{\mathrm{weak}} \equiv (m_X/g_X^2)g'^2 = 100$ GeV (upper dashed) and 1 TeV (lower dashed). The WIMPless hidden models generalize the WIMP miracle to a family of models with other dark matter masses and couplings [101, 102].

# Acknowledgements

It is a pleasure to thank the organizers Marco Cirelli, Babette Döbrich, and Jure Zupan for their support and extraordinary patience; the students of the 2021 Les Houches Summer School on Dark Matter for their interest and many questions; and Maximilian Dichtl and the anonymous reviewers for comments that have helped improve these notes. This work is supported in part by U.S. National Science Foundation Grants PHY-1915005, PHY-2111427, and PHY-2210283, Simons Investigator Award #376204, Simons Foundation Grant 623683, and Heising-Simons Foundation Grants 2019-1179 and 2020-1840.

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
