# Peer review of "The WIMP Paradigm: Theme and Variations"

_SciPost Physics Lecture Notes, doi:SciPost Phys. Lect. Notes 71 (2023)_

## Round 1 · Referee Report · Anonymous · 2022-12-31

Strengths

1) Well-organized pedagogical approach.
2) Proper balance of rigorous derivation and intuitive motivation.

Weaknesses

No weaknesses.

Report

These lecture notes provide and excellent discussion of the WIMP paradigm, WIMP searches, and non-standard variations on the WIMP paradigm. The discussion of the discrete WIMP Miracle is especially interesting, since, as the author notes, this aspect is underappreciated. The lectures strike the right balance between rigorous derivation and intuitive reasoning. The format and style are appropriate. These lecture notes meet the acceptance standards of the journal.

Requested changes

No changes needed.

---

## Round 1 · Referee Report · Anonymous · 2023-1-7

Report

This is set of lecture notes on WIMP dark matter based on lectures presented by Jonathan Feng at the 2021 Les Houches Summer School. The author is a leading figure in the field, and his own research and thinking on WIMPs has been influential. The subject is presented in an informal and lively manner. It begins by passing down some valuable general advice from our elders to the next generation of researchers, which sets a motivational tone. Much of the material covered is by now standard, including the motivation for WIMPs and the "WIMP miracle", canonical WIMPs in, e.g., supersymmetric models, and standard methods of WIMP detection. However, as the title suggests, connections are also made to close relatives of the WIMP, e.g., inelastic dark matter, WIMPless miracle, etc. It also contains insights that may be somewhat less appreciated today, such as the various motivations for symmetries that can serve a dual purpose of stabilizing WIMPs. At just 30 pages, this set provides a concise and engaging overview of WIMP dark matter which may be read before delving deeper into various technical aspects of Boltzmann equations, direct detection rates, and so on. In summary, this is an excellent and accessible introduction to WIMPs, which nicely complements the other topics in this Les Houches volume. It certainly meets and the standards of SciPost and should be published.

---

## Round 1 · Referee Report · Anonymous · 2023-1-11

Strengths

Very comprehensive introduction into the topic of WIMP dark matter, excellently written.

Weaknesses

- Text overlap with Ref 13 (arXiv:1003.0904) published in Ann. Rev. Astron. Astrophys. by the author of these Lecture Notes.
- Some of the discussed SUSY scenarios are outdated in view of the SM-like Higgs boson with mass of 125 GeV.

Report

These lecture notes are a write-up of Jonathan Feng's lectures on WIMP dark matter at the 2021 Les Houches Summer School, providing a very comprehensive, pedagogical introduction for newcomers to the field. The acceptance criteria (1. subject of ongoing interest to the research community and 2. a correct, systematic and intelligible presentation of the material) are definitely met.

Condensing the vast subject of WIMP dark matter to 36 pages is a difficult undertaking. Clearly, choices had to be made in order to stay concise, and overall this was done in an excellent manner. Nonetheless some additional details might be helpful in some places:
- On page 9, it is mentioned that chemical decoupling is distinct from kinetic decoupling ; a definition of kinetic decoupling and a reference for further reading would be helpful.
- Page 12, last paragraph of section 3.1: the argument that sneutrinos are typically not good dark matter candidates holds for left-sneutrinos, but not necessarily for right-sneutrinos, see Ref [32]; it would be nice to specify this.
- The CMSSM example, in particular the regions of parameter space shown in Figure 11 ($A_0=0$), is in conflict with the observed Higgs boson mass at 125 GeV; a clarifying comment would be in order.
- Last paragraph on page 16: there are many more ways of relaxing the minimality assumptions: adding a singlet (NMSSM), introducing Dirac instead of Majorana gauginos, adding right-handed neutrino superfields (with and w/o see-saw) etc etc.
- On page 21, the limit of EFT validity would merit a reference.
- Section 5.2: it would be instructive to have an explanation of the origin of isospin violation, or at least a reference to that end (e.g., Gao, Kang, Li, 1107.3529).
- Section 5.4: the minimal GMSB scenario has again a problem with $m_h=125$ GeV; this should at least be mentioned for clarity. After all, the mass of the SM-like Higgs is a prediction, not a free parameter, in these models.

Requested changes

1. The text overlap with Ref 13 should be mitigated or at least pointed out were relevant (or commented in the Introduction).
2. Consider the additional details mentioned in the report (optional apart from the issue of obtaining $m_h=125$ GeV in SUSY models, which should really be pointed out).

---

## Round 1 · Referee Report · Anonymous · 2023-2-4

Strengths

Excellent introduction to WIMP dark matter.
Strikes good balance of conceptual explanations and more in-depth calculations, equations, etc.

Weaknesses

I think a (brief) discussion of the Galactic Center Excess is warranted, especially since the DAMA signal is described in the direct detection sections.

Report

I would recommend publication after a few minor modifications.

Requested changes

1. Although not required, I think a sentence or two about how left-right mixing provides another way to enhance the annihilation of bino-like dark matter would be a good change either at the end of the last sentence on pg. 14 (the sentence just before Section 3.4), or possibly during the stau coannihlation discussion on pg. 16 in Section 3.5. Some possible references are 1707.02460, 2203.08107, 2209.13128 .
2. I think the claim that the bulk region is excluded is only true if there is no left-right mixing, see for example: 1406.4903 .
3. There seems to be an errant comma in Eq. 17
4. Some mention of the different neutrino fogs for different operators might be relevant to give: i.e. 1607.01468.
5. I think the Figure 12 caption should have more explanation as to what each of the lines and shaded regions represent for someone who is not necessarily familiar with these types of plots.
6. I think a sentence should be added explaining that the dark matter direct detection signal is expected to have an annual modulation due to the motion of the earth around the solar system at the beginning of the DAMA paragraph.
7. The sentence that begins “Photons point back to their source” needs to be changed to something similar to: “Since gamma ray photons have such high energies, they travel mostly undeflected and thus point back to their …”
8. In the last paragraph in 4.2 change "photons" to gamma ray photons.
9. As noted above, I think the Galactic center excess deserves some discussion. I would have expected it to have a similar level of discussion as DAMA signal, so maybe a paragraph. But I think at minimum it needs to be mentioned, and maybe a reference to a recent review could be given.

---

## Round 2 · List of Changes

** I thank all 5 reviewers for their comments, and I have added Maximilian and the anonymous reviewers to the acknowledgements. All comments are addressed below. I have modified the draft where I think it can be improved without adding too much to the length; details of the revisions are below. References have also been updated with publication data.

LIST OF CHANGES

REPORT 0

From: Maximilian Dichtl maximilian.dichtl@lpthe.jussieu.fr Subject: Re: Les Houches lecture notes by Feng Date: 23 December 2022 at 15:43:15 CET To: Marco Cirelli marco.cirelli@gmail.com

I want to make some general comments: The amount of effort put into this manuscript is very very high, so I found only two minor spelling mistakes, and also the structure of the paper is well thought out and very instructive, so there is no need to rearrange any paragraphs. The beginning is maybe written a bit too "flowery" for my personal(!!!) taste, but in a few places later-on in the text this kind of writing style (used sparingly) makes for a very entertaining read. I think that the lecture was one of the top 5 in Les Houches, and the notes have the same quality!

=== For Jonathan === You use two different phrases for defining the WIMP miracle: 1) motivated by cosmology, particle theory, and experiment, or 2) parameters (g,m) are that of the weak scale. I would like to see a more clear definition of the miracle per se, i.e. are 1) and 2) equivalent, or does 1) imply 2), or ...

I am aware that you don't want to go too much into mathematical details. Nevertheless, in the first half I pointed out some equations, where a short explanation on the mathematical steps performed can be added with very low effort but large learning effect.

In the chapter related to SUSY I made some comments formulated as questions. This is because I have never worked on SUSY, so these questions are coming from serious curiosity, but may seem totally naive to the expert, and the answers are trivial. Nevertheless, I'd be happy to see one or two sentences answering them in the paper, or a reference to further literature.

**These are replies to the handwritten notes in the pdf file I was sent. The line numbers refer to the version of the notes in that pdf file.

Line 89: The WIMP miracle is the fact that diverse considerations point toward a single region in (g,m) parameter space. I have clarified that WIMPs have (g, m) ~ (1, 10GeV-10TeV), and also added to the following footnote the fact that other particles with g<<1 and m<<10 GeV are also motivated by some of these considerations, but are not WIMPs. 128: I use \sim to indicate “of the order of.” 184, 188: Thank you: I added notes that I have assumed freezeout takes place during radiation domination and an adiabatically expanding universe. 203: addressed above. 297: the physical neutralinos are not necessarily Binos nor photinos; there is a complicated mixing. Explaining this in detail will take us too far afield. 331: added “for the simple calculation to make sense”. 353: fixed. 374: I trust readers to evaluate the range of mf/mW for themselves. 468: referred to [38, 39] and references therein. 502: keep as is (need cm^2 for comparison to the plot). 611: keep as is. 771: fixed. **781: keep as is (I mean that it is famous for being warm, not just famous!).

REPORT 1 Strengths 1) Well-organized pedagogical approach. 2) Proper balance of rigorous derivation and intuitive motivation. Weaknesses No weaknesses. Report These lecture notes provide and excellent discussion of the WIMP paradigm, WIMP searches, and non-standard variations on the WIMP paradigm. The discussion of the discrete WIMP Miracle is especially interesting, since, as the author notes, this aspect is underappreciated. The lectures strike the right balance between rigorous derivation and intuitive reasoning. The format and style are appropriate. These lecture notes meet the acceptance standards of the journal. Requested changes No changes needed. • Validity: Top • Significance: High • Originality: Good • Clarity: Top • Formatting: Excellent • Grammar: Excellent

**I thank the reviewer for the report; there are no comments to address.

REPORT 2 Report This is set of lecture notes on WIMP dark matter based on lectures presented by Jonathan Feng at the 2021 Les Houches Summer School. The author is a leading figure in the field, and his own research and thinking on WIMPs has been influential. The subject is presented in an informal and lively manner. It begins by passing down some valuable general advice from our elders to the next generation of researchers, which sets a motivational tone. Much of the material covered is by now standard, including the motivation for WIMPs and the "WIMP miracle", canonical WIMPs in, e.g., supersymmetric models, and standard methods of WIMP detection. However, as the title suggests, connections are also made to close relatives of the WIMP, e.g., inelastic dark matter, WIMPless miracle, etc. It also contains insights that may be somewhat less appreciated today, such as the various motivations for symmetries that can serve a dual purpose of stabilizing WIMPs. At just 30 pages, this set provides a concise and engaging overview of WIMP dark matter which may be read before delving deeper into various technical aspects of Boltzmann equations, direct detection rates, and so on. In summary, this is an excellent and accessible introduction to WIMPs, which nicely complements the other topics in this Les Houches volume. It certainly meets and the standards of SciPost and should be published. **I thank the reviewer for the report; there are no comments to address.

REPORT 3 Strengths Very comprehensive introduction into the topic of WIMP dark matter, excellently written. Weaknesses - Text overlap with Ref 13 (arXiv:1003.0904) published in Ann. Rev. Astron. Astrophys. by the author of these Lecture Notes. - Some of the discussed SUSY scenarios are outdated in view of the SM-like Higgs boson with mass of 125 GeV. Report These lecture notes are a write-up of Jonathan Feng's lectures on WIMP dark matter at the 2021 Les Houches Summer School, providing a very comprehensive, pedagogical introduction for newcomers to the field. The acceptance criteria (1. subject of ongoing interest to the research community and 2. a correct, systematic and intelligible presentation of the material) are definitely met. Condensing the vast subject of WIMP dark matter to 36 pages is a difficult undertaking. Clearly, choices had to be made in order to stay concise, and overall this was done in an excellent manner. Nonetheless some additional details might be helpful in some places: - On page 9, it is mentioned that chemical decoupling is distinct from kinetic decoupling ; a definition of kinetic decoupling and a reference for further reading would be helpful. **Added definition of kinetic decoupling.

  • Page 12, last paragraph of section 3.1: the argument that sneutrinos are typically not good dark matter candidates holds for left-sneutrinos, but not necessarily for right-sneutrinos, see Ref [32]; it would be nice to specify this. **Added clarifying footnote.

  • The CMSSM example, in particular the regions of parameter space shown in Figure 11 (A0=0), is in conflict with the observed Higgs boson mass at 125 GeV; a clarifying comment would be in order. **Added “In the particular slice of parameter space shown, the Higgs boson mass is typically lower than the measured value $m_h \simeq 125~\gev$, but \figref{relicSUSY} will serve well to illustrate some qualitative features.”

  • Last paragraph on page 16: there are many more ways of relaxing the minimality assumptions: adding a(with and w/o see-saw) etc etc. **Yes, hard to mention all possibilities, but I have added a mention of additional singlets.

  • On page 21, the limit of EFT validity would merit a reference.

**Added reference to Report of the ATLAS/CMS Dark Matter Forum [1507.00966].

  • Section 5.2: it would be instructive to have an explanation of the origin of isospin violation, or at least a reference to that end (e.g., Gao, Kang, Li, 1107.3529).

**Added this reference.

  • Section 5.4: the minimal GMSB scenario has again a problem with mh=125 GeV; this should at least be mentioned for clarity. After all, the mass of the SM-like Higgs is a prediction, not a free parameter, in these models.

**Added “GMSB models generically predict a Higgs boson lighter than the measured value, but this may be rectified by heavy superpartners~\cite{Feng:2012rn} or extra field content~\cite{Martin:2012dg}.”

Requested changes 1. The text overlap with Ref 13 should be mitigated or at least pointed out were relevant (or commented in the Introduction).
**Added to the intro that Ref. 13 serves as a source for these lectures.

  1. Consider the additional details mentioned in the report (optional apart from the issue of obtaining mh=125 GeV in SUSY models, which should really be pointed out).

**Done (see above).

• Validity: - • Significance: - • Originality: - • Clarity: - • Formatting: - • Grammar: -

REPORT 4 Strengths Excellent introduction to WIMP dark matter. Strikes good balance of conceptual explanations and more in-depth calculations, equations, etc. Weaknesses I think a (brief) discussion of the Galactic Center Excess is warranted, especially since the DAMA signal is described in the direct detection sections. Report I would recommend publication after a few minor modifications. Requested changes 1. Although not required, I think a sentence or two about how left-right mixing provides another way to enhance the annihilation of bino-like dark matter would be a good change either at the end of the last sentence on pg. 14 (the sentence just before Section 3.4), or possibly during the stau coannihlation discussion on pg. 16 in Section 3.5. Some possible references are 1707.02460, 2203.08107, 2209.13128 .

**Added this point and a reference to 1707.02460 to the end of Sec 3.4.

  1. I think the claim that the bulk region is excluded is only true if there is no left-right mixing, see for example: 1406.4903 .

**Added the suggested reference to Sec. 3.5.

  1. There seems to be an errant comma in Eq. 17

**Fixed – thanks.

  1. Some mention of the different neutrino fogs for different operators might be relevant to give: i.e. 1607.01468.

**Added the suggested ref where the neutrino fog is introduced.

  1. I think the Figure 12 caption should have more explanation as to what each of the lines and shaded regions represent for someone who is not necessarily familiar with these types of plots.

**Thank you. Added “The solid contours show the 90\% confidence limits from the experiments indicated. The green and yellow bands are the 1$\sigma$ and 2$\sigma$ sensitivity bands for LZ, and the dashed line shows the median of the LZ sensitivity projection.”

  1. I think a sentence should be added explaining that the dark matter direct detection signal is expected to have an annual modulation due to the motion of the earth around the solar system at the beginning of the DAMA paragraph.

**Added “The annual modulation signal arises from the motion of the Earth around the Sun, which results in greater scattering rates at certain times of the year.”

  1. The sentence that begins “Photons point back to their source” needs to be changed to something similar to: “Since gamma ray photons have such high energies, they travel mostly undeflected and thus point back to their …”

**Modified to “Since gamma ray photons have such high energies, they are typically not deflected and thus point back to their source, providing a powerful diagnostic.”

  1. In the last paragraph in 4.2 change "photons" to gamma ray photons.

**Done.

  1. As noted above, I think the Galactic center excess deserves some discussion. I would have expected it to have a similar level of discussion as DAMA signal, so maybe a paragraph. But I think at minimum it needs to be mentioned, and maybe a reference to a recent review could be given.

**Added: “A possible excess from a continuum signal from the galactic center has generated interest since its original observation~\cite{Goodenough:2009gk}.”

Validity: High Significance: Good Originality: Good Clarity: Top Formatting: Excellent Grammar: Excellent

---

## Editorial Decision

published